# miR-7 controls glutamatergic transmission and neuronal connectivity in a Cdr1as-dependent manner

Cledi A Cerda-Jara [1], Seung Joon Kim [1], Gwendolin Thomas[1], Zohreh Farsi[2], Grygoriy Zolotarov [1], Giuliana Dube[1], Aylina Deter[1], Ella Bahry [3], Elisabeth Georgii[4], Andrew Woehler[2], Monika Piwecka[5] & Nikolaus Rajewsky [1]✉

## Abstract

The circular RNA (circRNA) Cdr1as is conserved across mammals and highly expressed in neurons, where it directly interacts with microRNA miR-7. However, the biological function of this interaction is unknown. Here, using primary cortical murine neurons, we demonstrate that stimulating neurons by sustained depolarization rapidly induces two-fold transcriptional upregulation of Cdr1as and strong post-transcriptional stabilization of miR-7. Cdr1as loss causes doubling of glutamate release from stimulated synapses and increased frequency and duration of local neuronal bursts. Moreover, the periodicity of neuronal networks increases, and synchronicity is impaired. Strikingly, these effects are reverted by sustained expression of miR-7, which also clears Cdr1as molecules from neuronal projections. Consistently, without Cdr1as, transcriptomic changes caused by miR-7 overexpression are stronger (including miR-7-targets downregulation) and enriched in secretion/synaptic plasticity pathways. Altogether, our results suggest that in cortical neurons Cdr1as buffers miR-7 activity to control glutamatergic excitatory transmission and neuronal connectivity important for long-lasting synaptic adaptations.

**Keywords** circRNA; miRNA; Neuronal Activity; Cdr1as; miR-7
**Subject Categories** Neuroscience; RNA Biology

## Introduction

For about 10 years, it has been known that the circular RNA Cdr1as, originally discovered by the Kjems lab (Hansen et al, 2011), is one of the most highly expressed, vertebrate-conserved, circular RNAs in the mammalian brain (Memczak et al, 2013). Moreover, the expression of the microRNA miR-7 is deeply conserved, and one of its most frequent direct targets (out of all coding or noncoding RNAs) in the mammalian brain is Cdr1as, which is

specifically and highly expressed in excitatory neurons across the forebrain and harbors more than hundreds of highly conserved miR-7 binding sites (Hansen et al, 2013; Memczak et al, 2013; Piwecka et al, 2017; Rybak-Wolf et al, 2015). But what is the function of this unusual (no other known circular RNA has this many conserved binding sites for a specific miRNA) and deeply conserved interaction between two noncoding RNAs?

miR-7 is an ancient bilaterian miRNA, evolutionarily conserved across vertebrates and considered to be a prototypical neuroendocrine miRNA (Kredo-Russo et al, 2012; Latreille et al, 2014). Extensive comparative data indicate that miR-7 evolved within the neurosecretory brain (Christodoulou et al, 2010), and that the neuronal and pancreatic differentiation lineages are closely related (Zhao et al 2007). Some roles of miR-7 in mammalian cortical development have been described, mainly by its interaction with target gene Pax6 in neuronal progenitors (Pollock et al, 2014; Zhang et al, 2018), but nothing is known about miR-7 role in post-mitotic cortical regions. Mature miR-7 is enriched in central nervous system neuroendocrine glands and pancreatic tissues (Bravo-Egana et al, 2008; Hsu et al, 2008; Landgraf et al, 2007). In mouse pancreas, miR-7 has been shown to function as a negative regulator of stimulus-dependent insulin secretion (Latreille et al, 2014; Xu et al, 2015). This mechanism seems to be conserved for insulin-producing cells across the animal kingdom (Agbu et al, 2019; Latreille et al, 2014). In view of these findings, we hypothesized that miR-7 might regulate the release of key neuronal transmitters and that Cdr1as acts to control miR-7 function in cortical neurons.

Because some circRNAs are induced in neurites by homeostatic plasticity stimulation (You et al, 2015), and as we show that miR-7 is only lowly expressed in unstimulated cortical neurons, we speculated that the Cdr1as-miR-7 system may function mainly in stimulated neurons.

To test these ideas, we performed sustained neuronal stimulations and sustained perturbation of miR-7 expression in Cdr1as wild-type (WT) knockout (KO) and primary cortical neurons to investigate the dynamic interplay between Cdr1as and miR-7 in excitatory neurotransmission and neuronal network activity.

[1]Laboratory for Systems Biology of Gene Regulatory Elements, Berlin Institute for Medical Systems Biology, Max Delbrück Center for Molecular Medicine, Hannoversche Str. 28, 10115 Berlin, Germany. [2]Light Microscopy Platform, Berlin Institute for Medical Systems Biology, Max Delbrück Center for Molecular Medicine, Hannoversche Str. 28, 10115 Berlin, Germany. [3]Helmholtz Imaging, Max-Delbrück-Center for Molecular Medicine, Berlin, Germany Hannoversche Str. 28, 10115 Berlin, Germany. [4]Helmholtz AI, Helmholtz Zentrum München, Ingolstädter Landstraße 1, D-85764 Neuherberg, Germany. [5]Department of Non-Coding RNAs, Institute of Bioorganic Chemistry, Polish Academy of Sciences, Noskowskiego 12/14, 61-704 Poznan, Poland. ✉E-mail: rajewsky@mdc-berlin.de

We show that Cdr1as acts as a buffer that controls dynamic miR-7 regulation of glutamate release from presynaptic terminals and that this regulation strongly modulates neuronal connectivity responsible for long-lasting adaptation of synaptic plasticity.

# Results

## Sustained neuronal depolarization transcriptionally induces Cdr1as and post-transcriptionally stabilizes miR-7

To test our hypothesis that miR-7 is involved in cortical neurotransmitter secretion in similar ways as it is participating in stimulus-regulated secretion in pancreatic cells and neurosecretory glands (LaPierre et al, 2022; Latreille et al, 2014) and because some circRNAs were shown to be modulated specifically by neuronal activity (You et al, 2015), we tested the response of Cdr1as and miR-7 to sustained neuronal depolarization in primary cortical neurons from mice (Fig. 1A). Elevated extracellular K+ generates a global neuronal depolarization. Therefore, extracellular KCl treatments are useful for elucidating signaling, transcriptional, and plasticity-related events associated with homeostasis, and long-term potentiation, which controls several processes, such as sensory stimuli, learning and memory (Rienecker et al, 2020). A fast, primary response of neuronal stimulation are activity-regulated immediate early genes (IEG), dependent on the duration of the stimulation (Bartel et al, 1989; Tyssowski et al, 2018). More precisely, IEG are well known to be the primary transcriptional, rapid, and transient response to neuronal activation and depolarization stimuli and these transcripts encode mostly transcription factors and DNA-binding proteins (Curran and Morgan, 1987; Herdegen and Leah, 1998; Lanahan and Worley, 1998; Tischmeyer and Grimm, 1999).

We pretreated primary neurons at day in vitro 13 (DIV13) with TTX, CNQX, and AP5 overnight ("Methods") to inhibit spontaneous neuronal responses generated by Na+-dependent depolarization (TTX) or NMDA and non-NMDA glutamate release (CNQX, AP5), respectively. On DIV14, the culture media was exchanged with media containing high extracellular concentrations of potassium (KCl 60 mM). This treatment was performed in a detailed time course of 5, 15, 30, or 60 min, and compared to the corresponding control media with the same osmolarity (NaCl) as the high potassium media (Fig. 1B–D). No major cell death was observed upon KCl treatment compared to control (80% live cells) (Fig. EV1A).

The observed consistent and highly significant upregulation of IEG in K+ treated neurons (including master neuronal activity regulators such as *Fos, Fosb, Jun, Junb, Egr2, Nr4a1, Nr4a2,* Fig. 1B, blue dots) is consistent with previously described gene expression changes for one hour of sustained K+ treatment (Bartel et al, 1989, p. 89; Rienecker et al, 2020). The expression of genes not expected to react to stimulation was indeed not affected, for example housekeeping genes (*Actb, Vinculin, Tubb5*), other transcription factors (*Klf13, Atf3, Zic1*) and cell-cycle related genes (*Dusp1, Dusp5, Dusp6, IGF1R*) (Fig. 1B, black and gray dots).

Together with higher expression of activity-dependent IEGs, we observed a robust and significant upregulation of Cdr1as starting from 15 min of depolarization and remains consistently upregulated until the last timepoint measured (60 min) (Fig. 1C, red labels). In fact, at 60 min of K+ stimulation, the absolute expression levels of Cdr1as were the highest observed across all genes tested,

and its upregulation is similar in strength to what we have seen for IEGs (Fig. 1B, red labels). Moreover, of all circRNAs that we detected by total RNA sequencing ("Methods"), Cdr1as was by far the most highly expressed and induced circRNA (Fig. EV1A,B).

Mature miR-7a, was also strongly upregulated starting from 15 min after K+ stimulation, compared to the control (Fig. 1C, yellow labels). No significant changes were observed in the expression level of mature miR-671, involved in Cdr1as turnover. (Fig. EV1C). On the contrary, lncRNA Cyrano, the main negative regulator of miR-7 expression, is only significantly upregulated by the 5 min pulse of K+ depolarization, but not significantly altered at any other timepoint analyzed (Fig. 1B, purple labels). This time course of neuronal depolarization demonstrated the tight regulatory dynamics of Cdr1as:miR-7:Cyrano and how it changes accordingly with the time of stimulation. Short depolarization (5 min) rapidly increased Cyrano and this maintains miR-7 stable. Nonetheless, longer neuronal activation, increase directly or indirectly the transcription of Cdr1as as the main contributor to miR-7 abundance.

Interestingly, the expression of none of the primary transcripts of miR-7a (pri-miR-7a-1, pri-miR-7a-2) changed significantly after K+ stimulation (Fig. 1D). This shows that the regulation of mature miR-7 occurred post-transcriptionally. We further probed the dependence on transcription by blocking transcription with an elongation inhibitor (DRB) before or without K+ treatment (Fig. 1E,F). For none of the genes, significant upregulation was observed. Cdr1as and Cyrano levels were unperturbed, as well as mature miR-7a levels were unchanged, likely explained by the known high stability of Cdr1as (Memczak et al, 2013) and of miR-7 bound to AGO (Bartel, 2018) (Fig. 1E). Downregulation was observed for some IEGs and primary miR-7a isoforms, probably due to their short half-life times (Figs. 1F and EV1D,E).

We also tested the response of the Cdr1as RNA network to the same sustained depolarization in primary astrocytes exposed to the neuronal culture (glial feeder layer), and we did not observe any response after K+ stimulation, with or without DRB pre-treatment, compared to the control. Interestingly, we did not observe any upregulation of Cdr1as, Cyrano, IEGs, pri- or mature miR-7 or of any other miRNAs tested (Appendix Fig. S1A–C), suggesting that Cdr1as and miR-7a activity-dependent regulation is part of a neuronal-specific depolarization response mechanism.

Our observations demonstrate that miR-7 is post-transcriptionally regulated in a likely neuronal-specific way, either by regulation of maturation of the miR-7 precursor or by stabilization via interaction with Cdr1as. Strikingly, while most IEGs upregulated by depolarization are transcription factors encoding nuclear proteins, Cdr1as and Cyrano are enriched in the neuronal cytoplasm (Appendix Fig. S1D) and broadly expressed in neurites (Fig. EV2A; Appendix Fig. S2), where they both bind mature miR-7 (Hansen et al, 2013; Kleaveland et al, 2018; Memczak et al, 2013; Piwecka et al, 2017).

## Cdr1as co-localize in distances within synaptic compartments and is induced by the disinhibition of synaptic activity

In addition, we tested if the neurite distribution of Cdr1as RNA co-localized with pre- and postsynaptic proteins (vGLuT1, Synapsin-1, Homer1, and PDS-95). We combined smFISH with immunofluorescence and measured the distances from each Cdr1as molecule center to its nearest protein of interest ("Methods"). We confirmed the proximity

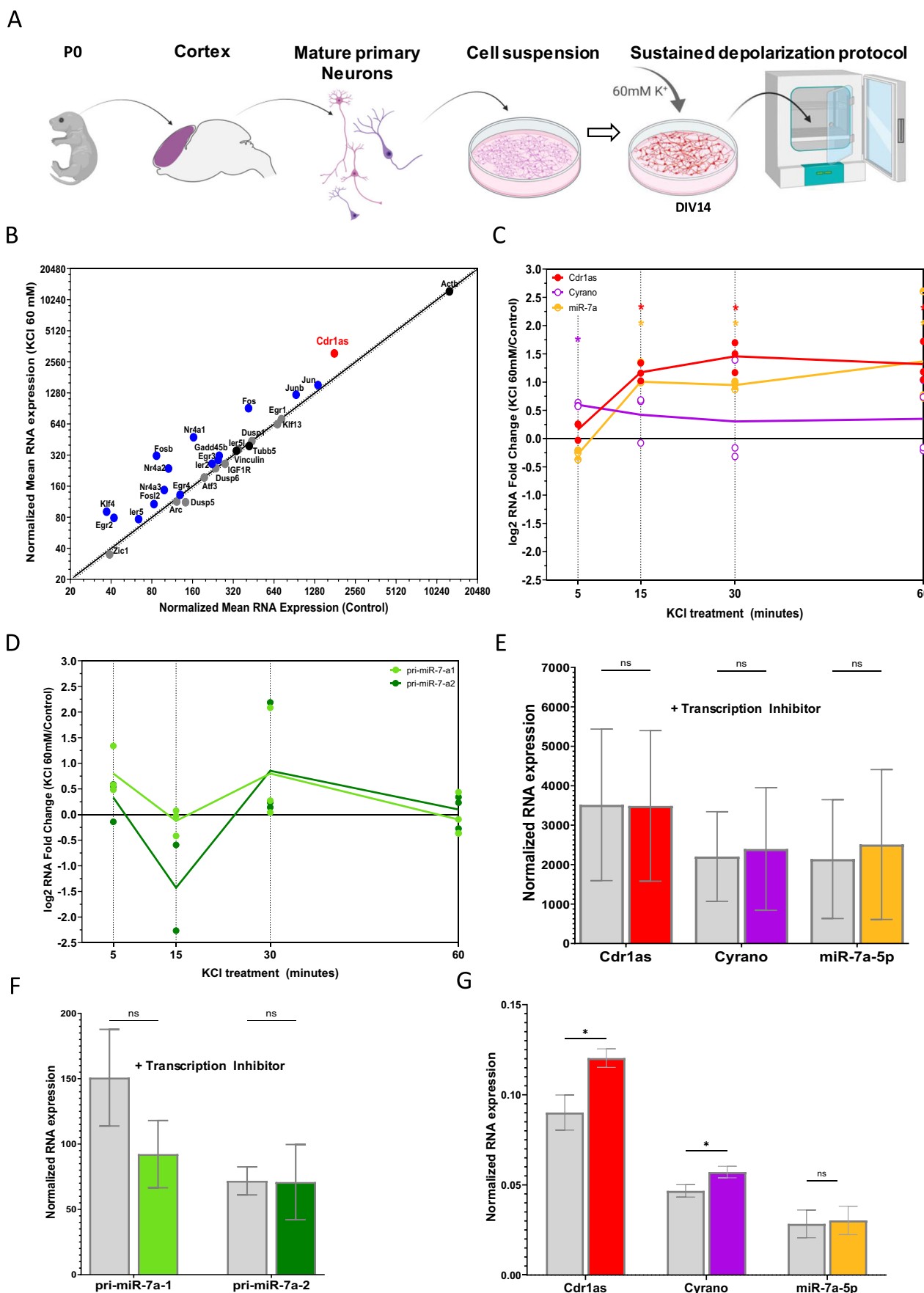

◀ **Figure 1.   Sustained neuronal depolarization causes transcriptional upregulation of Cdr1as and post-transcriptional stabilization of miR-7.**

(A) Protocol to culture primary cortical neurons (modified from Keach and Banker, 2006, "Methods"). DIV14 neurons, pretreated with TTX, CNQX, and AP5, were stimulated with KCl (60 mM) or corresponding osmotic control (NaCl) for 5, 15, 30, or 60 min. (B) RNA quantification after KCl depolarization (Nanostring nCounter, "Methods"). Statistically, significantly upregulated RNA levels are shown in blue (IEGs) and in red (Cdr1as). Housekeeping genes: black. RNA counts are normalized to housekeeping genes (*Actb, Tubb5 and Vinculin*). Each dot represents the mean of 4 biological replicates (four independent primary cultures from four animals). Black line: linear regression; dotted line: 95% confidence interval. (C) Cdr1as (red), miR-7a (yellow), and Cyrano (purple) RNA expression changes during time course of KCl depolarization (5, 15, 30, and 60 min), compared to corresponding osmotic control (log2FC). Cdr1as and Cyrano RNA measured by Nanostring nCounter ("Methods"). RNA counts are normalized to housekeeping genes (*Actb, Tubb5, and Vinculin*). Mature miR-7a quantified by TaqMan assay ("Methods"). Significant changes for each molecule are shown with asterisks in red (Cdr1as), yellow (miR-7a), and purple (Cyrano). Not statistically significant comparisons are not shown. Each dot represents one biological replicate (three independent primary cultures from three animals). Continuous line: mean value of the corresponding timepoint. *P* value: Mann–Whitney *U* test. (D) pri-miR-7a-1 (light green) and pri-miR-7a-2 (dark green) RNA expression changes during time course of KCl depolarization (5, 15, 30, and 60 min), compared to corresponding osmotic control (log2FC). RNA expression measured by Nanostring nCounter ("Methods"). RNA counts are normalized to housekeeping genes (Actb, Tubb5 and Vinculin). Each dot represents one biological replicate (three independent primary cultures from three animals). Continuous line: mean value of the corresponding timepoint. Not statistically significant comparisons are not shown. (E) Quantification of Cdr1as (red), Cyrano (purple) (Nanostring nCounter, "Methods") and miR-7a (yellow) (small RNA-seq, "Methods"), after KCl depolarization plus pre-incubation with transcription inhibitor (DRB), compared to corresponding osmotic control (gray bars). Nanostring RNA counts are normalized to housekeeping genes (*Actb, Tubb5, and Vinculin*), bar plot represents the mean of three biological replicates (three independent primary cultures from three animals). *P* value: Mann–Whitney *U* test. Error: SD. (F) Quantification of pri-miR-7a-1 (light green) and pri-miR-7a-2 (dark green) (Nanostring nCounter, "Methods") after KCl depolarization plus pre-incubation with transcription inhibitor (DRB), compared to corresponding osmotic control (gray bars). RNA counts are normalized to housekeeping genes (Actb, Tubb5, and Vinculin), bar plot represents the mean of three biological replicates (three independent primary cultures from three animals). *P* value: Mann–Whitney *U* test. Error: SD. (G) Cdr1as (red), miR-7a (yellow), and Cyrano (purple) RNA expression changes after 30 min incubation with 50 μM Bicuculline + 75 μM 4-Aminopyridine (antagonist of GABA-A receptors and K+ Channel blocker, respectively), compared to corresponding vehicle control (DMSO, gray bars). Cdr1as and Cyrano RNA measured by Nanostring nCounter ("Methods"). RNA counts are normalized to housekeeping genes (*Actb, Tubb5 and Vinculin*). Mature miR-7a quantified by TaqMan assay ("Methods"). The bar plot represents the mean of three biological replicates (three independent primary cultures from three animals). *P* value: Mann–Whitney *U* test. Error: SD. Source data are available online for this figure.

of Cdr1as with synaptic compartments. Colocalization of the circRNA with pre- and postsynaptic proteins has a median range of less than 3 μm (Appendix Fig. S2B–E). Being the closest distance with presynaptic vesicular glutamatergic transporter (vGluT1: 50% Cdr1as molecules ≤0.07 μm) (Appendix Fig. S2B–D) and postsynaptic density scaffolding protein Homer1(50% Cdr1as molecules ≤0.03 μm) (Appendix Fig. S2C–E). Therefore, we hypothesize that the cellular role of Cdr1as in neurons might be to safeguard or buffer the action of miR-7 within stimulated synapses (Appendix Fig. S2D,E).

Thus, to test the hypothesis of synaptic involvement of Cdr1as activity-dependent regulation, we performed an additional activity paradigm. We induced homeostatic synaptic plasticity by blocking inhibitory synapses of cortical neurons DIV14. We used a combination of Bicuculline and 4-AP ("Methods"), which allowed us to examine excitatory synaptic modulation of the cortical network. We observed a significant upregulation of Cdr1as and Cyrano, compared to the vehicle control, nevertheless, mature miR-7a was not significantly altered by Bicuculline stimulation, nor its primary transcripts (Figs. 1G and EV1F). This dynamic suggested that during synaptic stimulation, Cdr1as-dependent stabilization of mature miR-7a is counteracted by the upregulation of Cyrano (main negative regulator of miR-7 expression in neurons; Kleaveland et al, 2018). Similar to the effect we observed on brief KCl pulse (5 min), where Cyrano showed upregulation, but miR-7a is unaltered (Fig. 1B). Therefore, the primary stabilization role of Cdr1as on miR-7 might become relevant under stronger and long-term synaptic changes, where miR-7 levels are sustainably upregulated, as it could be for maintaining long-lasting plasticity potentiation.

## Synaptic terminals of Cdr1as-KO neurons show strongly increased presynaptic glutamate release after stimulation

To test if Cdr1as is involved in synaptic transmission in stimulated neurons, we used the cortical primary neurons of Cdr1as-KO animals (Piwecka et al, 2017) with their corresponding WT

littermates. We performed RNA quantification of marker genes at DIV21 and did not observe significant differences in the expression of neuronal (excitatory and inhibitory), glial, or proliferation and maturation markers between cultured Cdr1as-KO and WT cortical neurons (Fig. EV2C). In the cortex, glutamatergic neurons are the main cell type (Niciu et al, 2012), we corroborated that this is also true in our primary neuronal cultures from WT as well as Cdr1as-KO animals (Fig. EV2C). We then focused on measuring the activity of glutamate, as the main neurotransmitter relevant in our primary cultures.

To do so, we used a genetically-encoded glutamate sensor (GluSnFR; (Marvin et al, 2013) to perform real-time recordings of spontaneous and action potential-evoked (APs; electrical stimulation) glutamate release from individual presynaptic terminals in mature neurons DIV18 to 21, pretreated with TTX, CNQX, and AP5, to inhibit spontaneous or postsynaptic responses. Then, separately, we calculated the frequency of spontaneous neurotransmitter release over time (5 min of analysis) and the probability of evoked glutamate release in individual synapses during a sustained trend of electrical stimulations (20 APs at 0.5 Hz), as previously described (Farsi et al, 2021) (Fig. EV2D,E).

Our data showed a minor increase in spontaneous glutamate release in Cdr1as-KO presynaptic terminals as compared to WT neurons (Fig. 2A, pink Kernel-Density distribution), driven by the spontaneous activation of a group of very active synapses rather than by a global activation of all terminals (Fig. EV3A,B). This is consistent with previously shown upregulation of spontaneous miniature EPSC frequency in Cdr1as-KO hippocampal patch clamp experiments of single, isolated neurons (autapses, Piwecka et al, 2017). Strikingly, we observed a significant and stronger upregulation of action potential-evoked glutamate release in Cdr1as-KO neurons: 20 APs at 0.5 Hz for 40 s (Figs. 2B and EV3A,B). These observations suggest that Cdr1as might play a particularly important role in the modulation of excitatory transmission during sustained stimulation, in line with our hypothesis derived from K+ stimulation of WT neurons (Fig. 1).

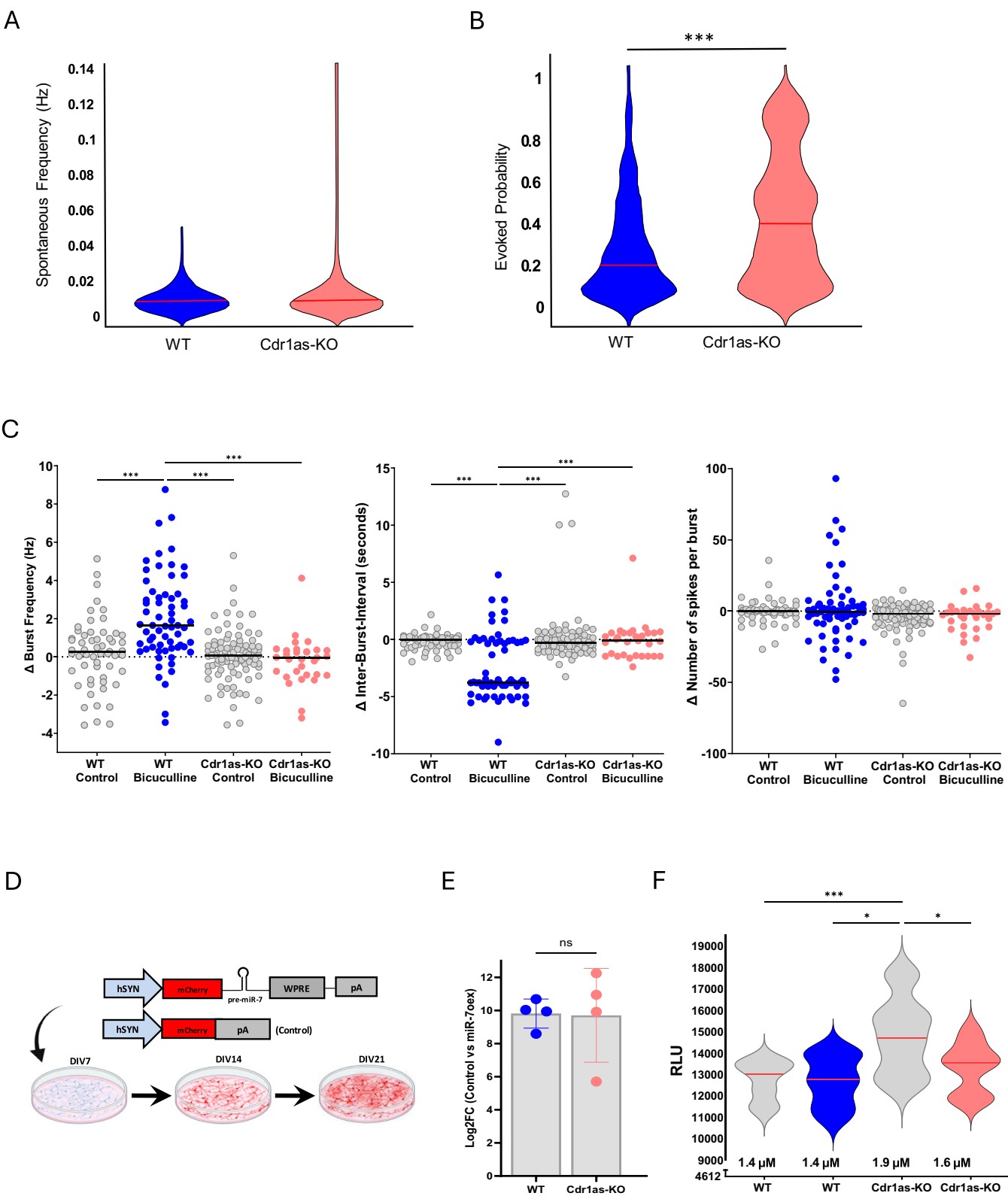

**Figure 2. Presynaptic terminals of Cdr1as-KO neurons show increased glutamate release and inhibition of synaptic adaptation.**

Glutamatergic phenotype is reverted by sustained miR-7 overexpression. (A) Quantification of spontaneous glutamate release calculated as spontaneous frequency. Violin plots: integration of the synapses from all animals used (WT; $n = 4$ animals and 662 synapses; Cdr1as-KO: $n = 3$ animals and 421 synapses; "Methods"). Synapses with fluorescence value $= 0$ were removed. Kernel-Density Estimate plotted ("Methods"). Red line: median. $P$ value: Mann–Whitney $U$ test. (B) Quantification of AP-evoked glutamate release calculated as evoked probability. Violin plots: integration of the synapses from all animals used (WT: $n = 4$ animals and 335 synapses; Cdr1as-KO: $n = 3$ animals and 270 synapses; "Methods"). Synapses with fluorescence value $= 0$ were removed. Kernel-Density Estimate plotted ("Methods"). Red line: median. $P$ value: Mann–Whitney $U$ test; ***<0.001. (C) Left panel: Burst Frequency—total number of single-electrode bursts divided by the duration of the analysis (600 s), in Hz. Middle panel: IBI—time in seconds between bursts averaged over the duration of the analysis. Right panel: Number of spikes of single-electrode bursts averaged over the duration of the analysis. Each dot represents a single-electrode recording from two independent primary cultures (WT Control = 57; WT bicuculline = 64; Cdr1as-KO Control = 92; Cdr1as-KO Bicuculline = 31 electrodes). Horizontal line: median. $P$ value: one-way ANOVA; ***<0.001. Multiple comparison correction test: Bonferroni. All not statistically significant comparisons are not shown. (D) Schematic representation of the miR-7a overexpression and the mCherry control constructs. Viral transduction performed at DIV7 (AAV, "Methods"). mCherry became visible at DIV14 and experiments were performed at DIV21. (E) Quantification of miR-7a upregulation based on reads from overexpression construct found in mRNA-Seq of WT and Cdr1as-KO neurons at DIV21, 14 days post miR-7a overexpression ("Methods"). Bar plots represent the mean value of four independent biological replicates per genotype. $P$ value: $U$ Mann–Whitney; n.s = 0.12. Error Bars: standard deviation. (F) Glutamate secretion assay performed on neuronal media collected from DIV18-21 WT and Cdr1as-KO neurons infected with control or miR-7a overexpression construct. Violin plots represent the integration of the secreted glutamate concentration quantified in independent primary cultures (WT = 12; WT + miR-7 overexpression = 12; Cdr1as-KO = 12; Cdr1as-KO + mirR-7 overexpression = 24 independent replicates). Glutamate concentrations are calculated based on a glutamate titration curve ("Methods", Fig. EV4H). The baseline concentration of glutamate in control media without cells was 0.28 μM (4612 RLU). Red line: median. $P$ value: one-way ANOVA; *0.033, ***<0.001. Multiple comparison correction test: Bonferroni. All not statistically significant comparisons are not shown. Source data are available online for this figure.

Overall, real-time recording of presynaptic neurotransmitter release exposed a direct link between Cdr1as-dependent regulation of glutamatergic transmission in resting and particularly in stimulated neuronal states. This revealed the synaptic explanation of the intrinsic and persistent excited state of Cdr1as-KO neurons (Piwecka et al, 2017).

## Induction of excitatory transmission inhibits synaptic modulations of chronically excited Cdr1as-KO neurons

We showed that Cdr1as expression levels are modulated by Bicuculline, a GABA receptor antagonist (Fig. 1G), and that Cdr1-KO neurons are secreting persistently more glutamate than WT neurons (Fig. 2B). Thus, we tested how Bicuculline stimulation of glutamatergic transmission affected the "chronically excited" Cdr1as-KO neurons.

We used a multi-electrode array well-based system (MEA, Axion BioSystems) to record local real-time extracellular field APs (Fig. EV3C) of WT and Cdr1as-KO neurons DIV21 treated with 10 μM Bicuculline for 1 h.

We observed that for WT neurons Bicuculline treatment was enough to significantly elevate electrical activity, by increase in burst frequency and decrease in the time interval in between bursts (Fig. 2C, blue dots). Strikingly, regardless of the enhanced presynaptic glutamate release, Cdr1as-KO neurons did not respond to the synaptic modulation. Cdr1as-KO cells did not show any significant changes in bursts or spikes numbers upon bicuculline stimulation (Fig. 2C, pink dots). These results suggest that Cdr1as-KO neurons adapted a compensatory cellular mechanism to allow control of synaptic modulations. A counteract of the chronic presynaptic activation that therefore will keep network activity stable upon stimuli.

How could Cdr1as regulate glutamatergic transmission in order to modulate long-lasting synaptic plasticity? It has been proven that Cdr1as' main binding partner, miR-7, directly acts as a negative regulator of stimulus-regulated secretion in secretory glands, where miR-7 is within the top highly expressed miRNA (Bravo-Egana et al, 2008, p. 7; Landgraf et al, 2007; Latreille et al, 2014). However, cortical neurons are characterized by a very low basal expression of mature miR-7 when compared to characteristic neuronal miRNAs

(Fig. EV4A–C; Dataset EV1). Therefore, to reliably study the functions of miR-7 in cortical neurons and to molecularly mimic the increased miR-7 levels observed as a consequence of sustained neuronal stimulation ($K^+$ treatments; Fig. 1B,C), we decided to induce miR-7 expression in unstimulated WT and Cdr1as-KO neurons.

## Sustained miR-7 overexpression reverts increased glutamate secretion in Cd1as-KO neurons

We created a neuron-specific (human synapsin promoter; hSYN) long-term transgene overexpression system, consisting of a miR-7 precursor transcript (pre-miR-7a-1) coupled to a fluorescent reporter (mCherry) and packaged into an adeno-associated virus particle (AAV) for highly efficient neuron transduction, together with a fluorescent vector control (Fig. 2D, "Methods").

We infected WT and Cdr1as-KO primary neuronal cultures on DIV7 with miR-7 or control AAV particles and monitored mCherry as a reporter of miR-7 expression until DIV21 (Fig. 2D). We then tested miR-7 overexpression and confirmed significant induction of miR-7 in WT and Cdr1as-KO cultures. We obtained an upregulation of Mir-7-a-1 transcript of 1000-folds at the mRNA level (overexpression construct; Fig. 2E), which translated into a 32-fold increase of mature miR-7a-5p (Fig. EV4C). Importantly, induced miR-7 levels were equal in both genotypes (Figs. 2E and EV4C). In addition, we did not observe significant change of miR-671 or the highly expressed neuronal miRNA let-7a in any genotype (Fig. EV4F), indicating that our overexpression did not interfere with the abundance of other miRNAs.

We observed that the mCherry protein became visible starting from DIV14 onward (7 dpi). Using single-molecule RNA in situ hybridization of miR-7, we demonstrate that the overexpressed miRNA is found homogenously distributed in somas and neurites of neurons from both genotypes (Fig. EV4E), while in control neurons (not infected or empty control) the signal is sparse and only few molecules are captured (~40 copies per cell; Fig. EV4D). Accordingly, we kept DIV14 (7 dpi) as the starting timepoint to monitor neuronal activity in all following experiments.

We then measured secreted glutamate in the media of DIV18-21 WT and Cdr1as-KO neuronal cultures at resting state, in neurons infected with control or miR-7 overexpression AAV (14 dpi). For

this, we set up a bioluminescent enzymatic assay ("Methods"), based on the activity of glutamate dehydrogenase (GDH) coupled to the activity of Luciferase (Ultra-GloTM rLuciferasa, Promega), which results in the production of light proportional to the concentration of glutamate in the media (Fig. EV4G). Exact concentrations of secreted glutamate were calculated by interpolation of luminescence values (RLU) from a glutamate standard curve (Fig. EV4H).

Secreted glutamate levels were not changed in WT cells when overexpressing miR-7 (Fig. 2F, blue density distribution), in contrast to other systems such as pancreatic *beta* cells where specific overexpression of miR-7 induces a reduction of insulin secretion (Latreille et al, 2014). However, consistent with our observations (Fig. 2A,B), glutamate levels were strongly and significantly increased in Cdr1as-KO neurons (Fig. 2F, gray density distributions). Strikingly, this increase was reverted to almost WT levels when miR-7 was overexpressed in the absence of Cdr1as (Fig. 2F, pink density distribution). As Cdr1as is absent or only spuriously expressed in the pancreas, our data strongly suggests that Cdr1as in cortical neurons has evolved to buffer miR-7 function.

## Sustained miR-7 overexpression reverted increased local neuronal bursting in Cdr1as-KO neurons

To investigate more detailed neuronal activity differences associated with the loss of Cdr1as, along with examining if and how those differences are associated with miR-7, we recorded local and network real-time activity over neuronal maturation (DIV7 until DIV21), in WT and Cdr1as-KO cultures, with or without miR-7 overexpression. We used the multi-electrode array system (MEA, Axion BioSystems) to record neuronal spikes and ultimately network activity (Fig. EV3C). No batch effect was observed due to biological replicates (Fig. EV3D).

When measuring local APs recorded by an electrode placed nearby the activated neurons, we observed that throughout maturation time (and similar to our data for glutamate secretion in media), there are no significant changes in the frequency of firing in WT neurons overexpressing miR-7 as compared to WT controls in any of the analyzed timepoints (Figs. 3A; Appendix Fig. S3A, blue and gray–blue datapoints). However, Cdr1as-KO neurons consistently showed a significant upregulated spontaneous firing frequency (larger number of functional APs per second), which progressively increased from DIV7 until DIV21 as compared to WT neurons (Fig. 3A; Appendix Fig. S3A, pink and blue datapoints). Again, this upregulation was reverted to almost WT levels (or exceeding those) when overexpressing miR-7 in Cdr1as-KO neurons (Fig. 3A; Appendix Fig. S3A, pink and gray-pink datapoints). This "rescue" was the strongest in more mature neurons (DIV21). Together, we conclude that our observations of upregulated glutamate secretion in Cdr1as-KO neurons (Fig. 2) translate into increased functional APs.

To test if changes in firing rates are linked to changes in patterns of neuronal activity, we measure neuronal burst formation as defined as rapid AP spiking followed by inactive periods much longer than typical interspike intervals ("Methods"). In WT neurons no significant differences in bursting frequency were observed after sustained overexpression of miR-7 compared to controls, at any timepoint (Fig. 3B; Appendix Fig. S3B, blue and gray–blue).

However, Cdr1as-KO neurons bursting activation patterns were persistently increased over maturation. This increase in bursting was as well reverted by miR-7 overexpression to WT levels or even to lower frequencies (Fig. 3B; Appendix Fig. S3B, pink and gray-pink datapoints).

Cdr1as-KO neurons also showed significantly longer burst durations, (Fig. 3C; Appendix Fig. S3C, pink and blue datapoints) and a higher number of functional spikes per burst (Appendix Fig. S3D,E, pink and blue datapoints), both growing in deviation from WT as the neurons matured. Interestingly, sustained miR-7 overexpression only significantly affected burst duration in DIV21 neurons, and it did so in opposite ways, depending on the neuronal genotype, increasing the duration in WT neurons, and decreasing it in Cdr1as-KO.

Altogether, the real-time recording of local neuronal activity not only demonstrated that AP frequencies are impaired in Cdr1as-KO neurons, but suggested that cellular mechanisms responsible for transforming local AP into coordinated activation patterns (bursting activity) are affected in Cdr1as-KO neurons and all defects are effectively rescued by sustained mir-7 overexpression. Longer, more frequent, higher number of spikes per bursts are indications of stronger excitatory transmission, or of less inhibition, as it takes longer time to shut down a burst (Zeldenrust et al, 2018). Therefore, our local neuronal activity results suggest that the effect of the interaction of Cdr1as and miR-7 on glutamatergic release (Fig. 2) is linked to the modulation of some aspects of synaptic plasticity, neural coding, and network synchronization.

## Sustained miR-7 upregulation strongly affects neuronal connectivity in a Cdr1as-dependent manner

To investigate further how Cdr1as and miR-7 are relevant to the modulation of synaptic plasticity and neuronal network patterns, we explored more complex neuronal wiring parameters, taking advantage of multi-electrode simultaneous measurements.

First, we measured network synchrony by comparing the activity profiles of two neighboring electrodes across different time delays (area under the cross-correlation curve). When the area is 0 there is perfect synchrony, while larger area values indicate lower synchronicity. We observed a more asynchronous network in Cdr1as-KO neurons at late states of maturation (DIV14 to DIV21) when compared to WT (Figs. 3D, pink and blue datapoints and EV3E). Moreover, after sustained miR-7 overexpression the synchronicity was significantly increased for both genotypes. The size of this miR-7-dependent effect was much stronger in the case of Cdr1as-KO neurons (Fig. 3D; Appendix Fig. S3F, gray–blue and gray-pink datapoints). This higher synchrony could also be explained by reduced neuronal activity after miR-7 overexpression, as shown in our local neuronal activity observations (Fig. 3A). Second, we analyzed AP oscillations across the network and captured the distribution of action potentials.

We measured the coefficient of variation of the intervals interspike (ISI CoV), where zero (0) values indicate perfect Poisson distribution of APs and higher values represent irregular bursting. We observed significantly higher oscillation periodicity in Cdr1as-KO network at very late points of maturation (DIV18-DIV21). In WT and Cdr1as-KO neurons, after sustained miR-7 overexpression there was a significant downregulation of oscillatory bursting (Fig. 3E; Appendix Fig. S3G, gray–blue and gray-pink datapoints).

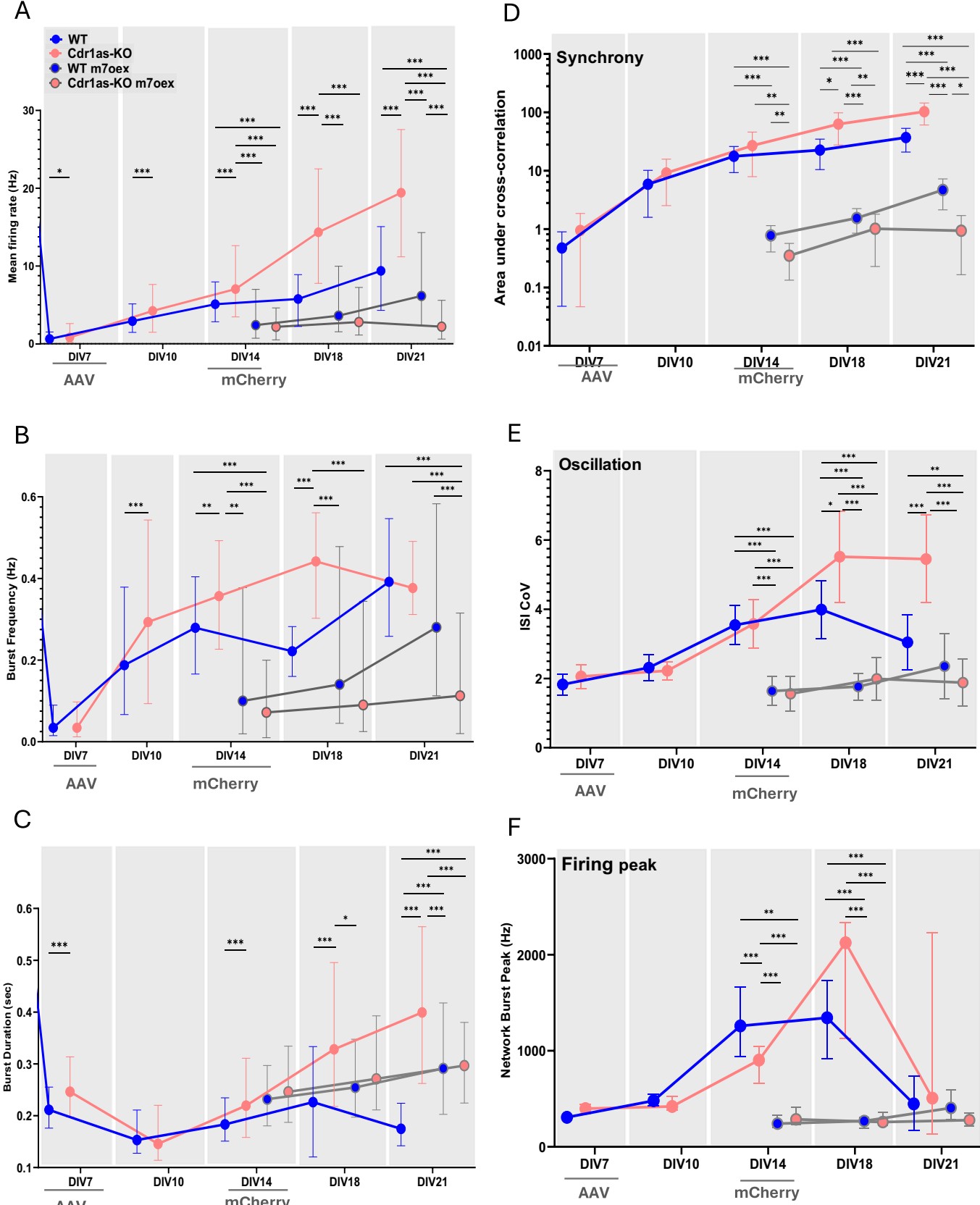

**Figure 3. Removal of Cdr1as causes increased frequency and duration of local action potentials, neuronal bursting, and affects neuronal connectivity by changing network synchrony and oscillation.**

Sustained miR-7 expression reverted these effects. (**A**) Mean Firing Rate: Total number of spikes per single-electrode divided by the duration of the analysis (600 s), in Hz. Median value across timepoints with correspondant SD is plotted. Each timepoint represents a single-electrode recording from four independent primary cultures (WT = 187; WT + miR-7 overexpression = 194; Cdr1as-KO = 189; Cdr1as-KO + miR-7 overexpression = 200 electrodes); transduction time (AAV, DIV7) and reporter first visualization (mCherry, DIV14) are indicated. *P* value: two-way ANOVA; *0.033, **0.002, ***<0.001. Multiple comparison correction test: Bonferroni, all other comparisons were not statistically significant (not shown). (**B**) Burst Frequency: Total number of single-electrode bursts divided by the duration of the analysis, in Hz. Panel plotted as in (**A**). (**C**) Burst Duration: Average time (sec) from the first spike to last spike in a single-electrode burst. Panel plotted as in (**A**). (**D**) Asynchrony: The ability of neurons to generate APs simultaneously was calculated as area under the well-wide pooled inter-electrode cross-correlation. Higher areas indicate lower synchrony (Halliday et al, 2006, "Methods"). Median value across timepoints with correspondant SD is plotted. Each timepoint represents a network recording ("Methods") from four independent primary cultures (WT = 11; WT + miR-7 overexpression = 26; Cdr1as-KO = 12; Cdr1as-KO + miR-7 overexpression = 28 electrodes); transduction time (AAV, DIV7) and reporter first visualization (mCherry, DIV14) are indicated. All metrics apply to network bursts across single wells within 20 ms. *P* value: two-way ANOVA; *0.033, **0.002, ***<0.001. Multiple comparison correction test: Bonferroni, all other comparisons were not statistically significant (not shown). (**E**) Oscillation: Average across network bursts of the interspike interval coefficient of variation (ISI CoV) (standard deviation/mean of the interspike interval) within network bursts. Oscillation is a measure of how the spikes from all of the neurons are organized in time. WT = 11–26; Cdr1as-KO: 12–28 independent network recordings. Panel plotted as in (**D**). (**F**) Burst Peak: Maximum number of spikes per second in the average network burst. The peak of the average network burst histogram divided by the histogram bin size to yield spikes per sec (Hz). WT = 11–26; Cdr1as-KO: 12–28 independent network recordings. Panel plotted as in (**D**). Source data are available online for this figure.

The magnitude of the miR-7-dependent effect in WT neurons was much smaller compared to Cdr1as-KO, and completely disappeared at DIV21. Network oscillation differences in Cdr1as-KO, which were efficiently modulated by the sustained high expression of miR-7, is a strong indication of an imbalance between glutamatergic and inhibitory signaling in Cdr1as-KO neuronal network.

Finally, to study timepoints where the network reaches its maximal firing peak (maturation of the network) we measured the maximum spike frequency in the average network burst. We did not find significant differences between WT and Cdr1as-KO neurons over time, and we observed a significant downregulation on the burst peak after miR-7 overexpression affecting equally both genotypes (Fig. 3F; Appendix Fig. S3H).

These results suggest that the modulation of miR-7 expression has a great influence on neuronal network activity regulation, but the extent and strength of this influence is modulated by the expression of Cdr1as. Therefore, in constitutive absence of Cdr1as the system is more prone to the effects of miR-7, especially noticeable in very late times of neuronal synaptic maturation.

## Sustained miR-7 overexpression downregulates Cdr1as and restricts its residual expression to the soma

To better understand molecular mechanisms underlying Cdr1as:-miR-7 interaction and potential interactions with lncRNA Cyrano, we performed single-molecule RNA in situ hybridization (smFISH Stellaris; "Methods") for Cdr1as and Cyrano (Fig. 4A,B; Appendix Fig. S4A,B), using Cdr1as-KO cells as negative control (Fig. EV2B).

Surprisingly, we noticed that upon sustained miR-7 overexpression in WT neurons, the distribution of Cdr1as molecules was restricted to neuronal somas and close proximal space, while massively cleared from all neurites (Fig. 4A,B). This is observable, for example, at the somato-neuritic areas (Fig. 4A,B, right zoom-in).

Accordingly, Cdr1as molecules were quantified across neuronal replicates in independent cultures from two different animals of control and miR-7 overexpression conditions ("Methods"). In the two independent animals quantified, we observed a significant reduction in Cdr1as molecules within neuronal somas. Similarly, a significant clearance of Cdr1as happened within the neurites

following the upregulation of miR-7. ($9.3 \times 10^{-5}$ molecules/μm. Figure 4C, somas; Fig. 4D, neurites).

We did not observe any evident change in Cyrano distribution after miR-7 overexpression (Appendix Fig. S4A,B). Nonetheless, the single-molecule quantification showed a significant reduction of Cyrano molecules in somas, but no differences within neurites Appendix Fig. S4C,D.

An independent direct quantification of RNA expression using the Nanostring system ("Methods") confirmed statistically significant decrease of Cdr1as (Fig. 4E, blue dots) in WT neurons after miR-7 overexpression, but not significant global downregulation of Cyrano in the same neurons (Fig. 4F, blue dots). Interestingly, in Cdr1as-KO neurons we did observe Cyrano RNA expression significantly reduced after sustained miR-7 overexpression (Fig. 4F, pink dots). This suggests that the strength of the negative regulation of Cyrano expression, resulting from sustained miR-7 activity, may depend on the expression levels of Cdr1as.

To investigate if Cdr1as and Cyrano are spatially co-localized in control conditions or after miR-7 overexpression, we used a computational pipeline (Eliscovich et al, 2017) for quantification of single-molecule spatial correlations ("Methods"). We measured all mutual nearest neighbor distances (MNN) of Cdr1as-Cyrano molecule associations in comparison with positive and negative controls ("Methods"). The results from the cumulative distribution of MNN distances between each pair of probes showed no significant association in space between Cdr1as and Cyrano molecules neither in control neurons nor after miR-7 overexpression, as compared to positive controls (Fig. EV5A). The association in space was tested in whole cells, neurites, and somas, separately (Fig. EV5B). We did not find significant differences between Cdr1as-Cyrano distances and the negative control in any of the tested compartments, for either of the two tested experimental conditions. Therefore, we conclude that Cdr1as and Cyrano are not specifically co-localized in a biologically relevant distance.

Altogether, these data indicate that we have identified a new layer of regulation of the Cdr1as-Cyrano-miR-7 axis where not only expression change of the lncRNAs affect miR-7 levels, as reported by others before (Kleaveland et al, 2018; Piwecka et al, 2017).

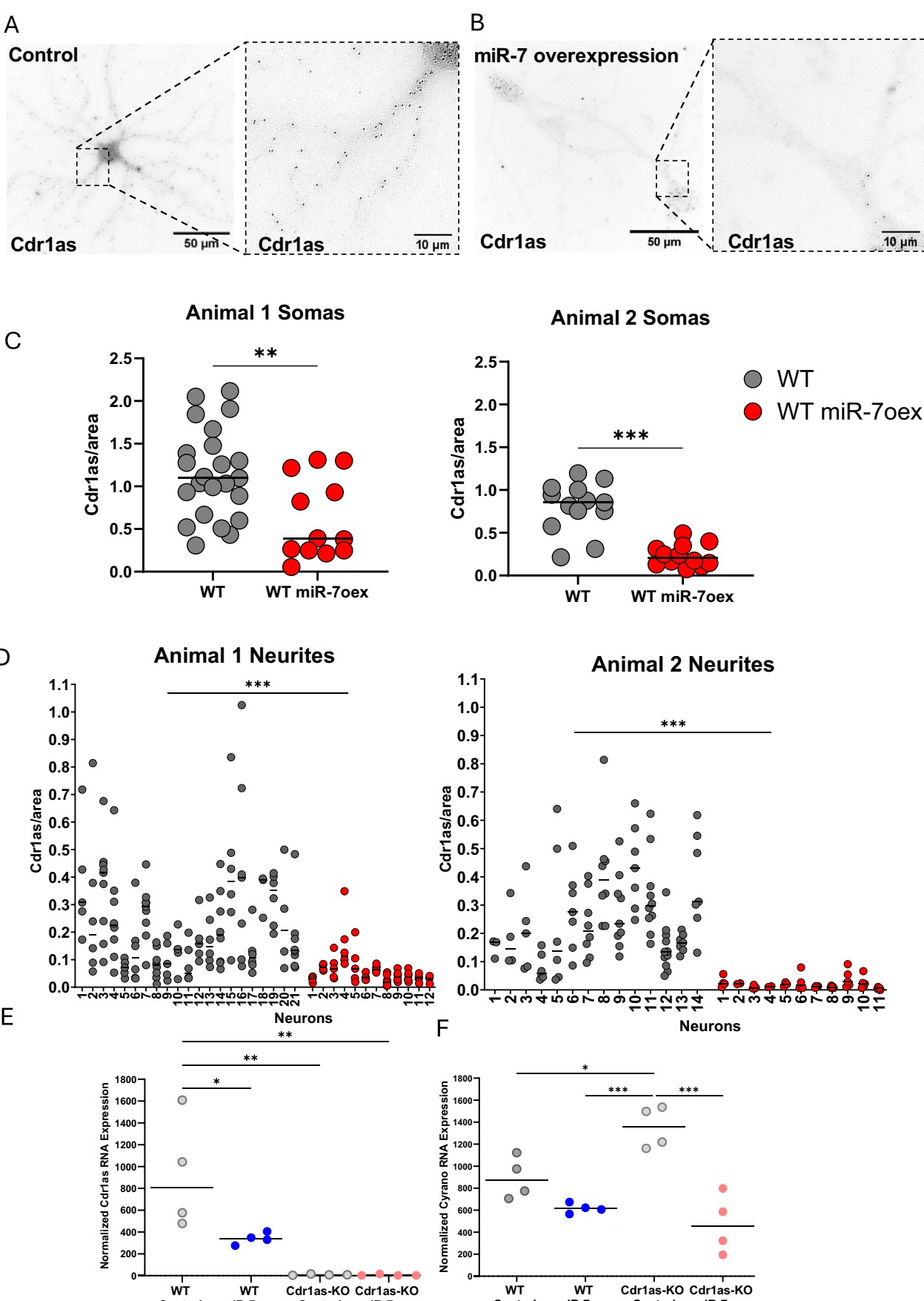

**Figure 4. miR-7 overexpression downregulates Cdr1as, removes it from neurites, and restricts its residual expression to the soma.**

(A) Single-molecule RNA FISH (Stellaris, "Methods") of Cdr1as (black dots) performed in WT neurons DIV21 from control condition. Left: single neuron at 60X. Right: zoom-in of somato-neuritic area. (B) Single-molecule RNA FISH (Stellaris, "Methods") of Cdr1as (black dots) performed in WT neurons DIV21 infected with control or miR-7a overexpression construct. Left: neurons imaged at 60X. Right: zoom-in of somato-neuritic area. (C) smRNA FISH quantification of Cdr1as molecules in somas, separated by biological replicates (following Raj et al, 2008, "Methods"). Each dot represents the mean number of Cdr1as molecules in an independent soma normalized by area (100 px = 21.5 μm). Left: animal 1; Control neurons: 23 somas, miR-7 overexpression neurons: 13 somas. Right: animal 2; Control neurons: 13 somas, miR-7 overexpression neurons: 12 somas. Horizontal lines: Median. *P* value: *U* Mann–Whitney test (nonparametric, unpaired); **0.002, ***<0.001. (D) smRNA FISH quantification of Cdr1as molecules in neurites, separated by biological replicates (following Raj et al, 2008, "Methods"). Each dot represents the mean number of Cdr1as molecules in an independent neurite normalized by area (100 px = 21.5 μm). Left: animal 1; Control neurons: 132 neurites (from 21 neurons), miR-7 overexpression neurons: 62 neurites (from 12 neurons). Right: animal 2; Control neurons: 98 neurites (from 14 neurons), miR-7 overexpression neurons: 43 neurites (from 11 neurons). Horizontal lines: Median. *P* value: *U* Mann–Whitney test (nonparametric, unpaired); ***<0.001. (E) Quantification of Cdr1as (Nanostring nCounter, "Methods"), performed in WT and Cdr1as-KO neurons DIV21 from neurons infected with control or miR-7a overexpression construct. RNA counts are normalized to housekeeping genes (*Actb, Tubb5,* and *Vinculin*). Each dot represents an independent biological replicate (four independent primary cultures from four animals). *P* value: one-way ANOVA; *0.033, **0.002. Multiple comparison correction test: Bonferroni. Horizontal bar: Median. All other comparisons were not statistically significant (not shown). (F) Quantification of Cyrano (Nanostring nCounter, "Methods"), performed in WT and Cdr1as-KO neurons DIV21 from neurons infected with control or miR-7a overexpression construct. RNA counts are normalized to housekeeping genes (*Actb, Tubb5* and *Vinculin*). Each dot represents an independent biological replicate (four independent primary cultures from four animals). *P* value: One-way ANOVA; *0.033, ***<0.001. Multiple comparison correction test: Bonferroni. Horizontal bar: Median. All other comparisons were not statistically significant (not shown). Source data are available online for this figure.

We here show that exposure to sustained high concentrations of miR-7 generates a negative feedback loop that can downregulates Cdr1as in WT conditions from both somas and neurites (Fig. 4A–E). The impact in the regulation of Cyrano depends on the presence of Cdr1as (Fig. 4F; Appendix Fig. S4).

The observations regarding Cdr1as decreased in WT neurons (Fig. 4A–D), may explain in part the reduced phenotype observed in local and network neuronal activity, as well as glutamate secretion of WT neurons after sustained miR-7 overexpression (Figs. 2F and 3A–F). A gradual yet global diminution of miR-7's principal regulator (Cdr1as) within the intracellular compartments of WT neurons over the temporal course of miR-7 overexpression, may be linked to the emergence of a functional phenotype similar to that of Cdr1as-KO neurons. This implies that WT neurons might manifest phenotypes akin to those indicative of an indirect acute knockdown of miR-7 regulators.

## Transcriptomic changes caused by sustained miR-7 overexpression are enhanced by the absence of Cdr1as

We further investigated whether molecular consequences of the interaction of Cdr1as and miR-7 are consistent with the defects observed on cortical neuronal functions (glutamate release and electrophysiological defects). We performed bulk mRNA sequencing of four independent primary neuronal cultures at DIV21 from WT and Cdr1as-KO animals, infected with control or miR-7 overexpression (14 dpi) and decipher molecular signatures. Our identification of differentially expressed genes was corrected for batch effects and individual effects ("Methods", Fig. EV5C).

First, we investigated predicted miR-7 target genes ("Methods"). We did not find statistically significant global expression changes of computationally predicted miR-7 target genes when comparing WT and Cdr1as-KO control neurons (Fig. 5A, left panel; Dataset EV2). However, for both, WT and Cdr1as-KO neurons, sustained miR-7 upregulation, resulted in a significant global expression decrease of miR-7 targets when compared to non-target RNA transcripts. Even more, we noticed that in the case of Cdr1as-KO neurons, miR-7 predicted targets were clearly more strongly downregulated as compared to WT neurons in the same conditions (significant difference between the shift of cumulative fractions: *P* value < 1e10$^{-6}$; Fig. 5A, middle and right panel; Fig. EV5D, purple and blue lines; Dataset EV2).

Because miR-7 sustained upregulation was equally efficient in both genotypes (Figs. 2E and EV4C–E), our miR-7 targets data suggest that the interaction between Cdr1as and miR-7 is responsible for the differences in observed miR-7 targets regulation strength. This indicates that the constitutive loss of Cdr1as is sensitizing the neurons to the action of miR-7 on its mRNA targets. Predicted targets from a control miRNA not expressed in brain were not altered in any of the tested comparisons (miR-122-5p; Fig. EV5E). Can we directly link some specific miR-7 target regulations to the efficient rescue of glutamate release observed in Cdr1as-KO neurons? Several predicted miR-7 target genes have been experimentally validated as direct miR-7 targets that operate as regulators of secretion in pancreatic cells and recently in hypothalamus, both of them thoroughly studied endocrine cell tissues with= high levels of miR-7 expression (Bravo-Egana et al, 2008; Kredo-Russo et al, 2012; Landgraf et al, 2007; LaPierre et al, 2022; LaPierre and Stoffel, 2017). Out of these validated miR-7 target genes, *Snca*, a gene encoding for a presynaptic scaffold protein and involved in many neurodegenerative diseases (Bennett, 2005), was the most downregulated miR-7 target gene after miR-7 overexpression regardless of Cdr1as expression (Fig. 5B). We also observed that most of these experimentally validated miR-7 targets that function in pancreatic insulin secretion (Latreille et al, 2014) are exclusively and significantly downregulated in Cdr1as-KO neurons after sustained miR-7 overexpression (Fig. 6B). Interesting examples include central regulator of vesicle fusion and SNARE activity (*Cplx1*), cytoskeleton remodeler (*Pfn2*) and an enzyme responsible for the regulation of dendritic growth and inhibitory synapses formation (*Zdhhc9*) (Fig. 5B, right panel).

Together these observations suggest that the role of miR-7 as a regulator of secretion is conserved between secretory glands (i.e., insulin secretion) and cortical neurons (glutamate secretion), with the crucial difference that the strength of miR-7 action in cortical neurons depends on Cdr1as levels.

Besides the strong dependence of miR-7 target genes regulation on Cdr1as expression, our transcriptomic data revealed many other genes which are dysregulated upon Cdr1as and/or miR-7 perturbation (Fig. 5C). Of course, indirect effects are to be expected, but the vast increase in dysregulated genes when combining miR-7 overexpression in Cdr1as-KO neurons points at a synergetic regulation exerted by both RNAs (Fig. 5C, right panel; Dataset EV3).

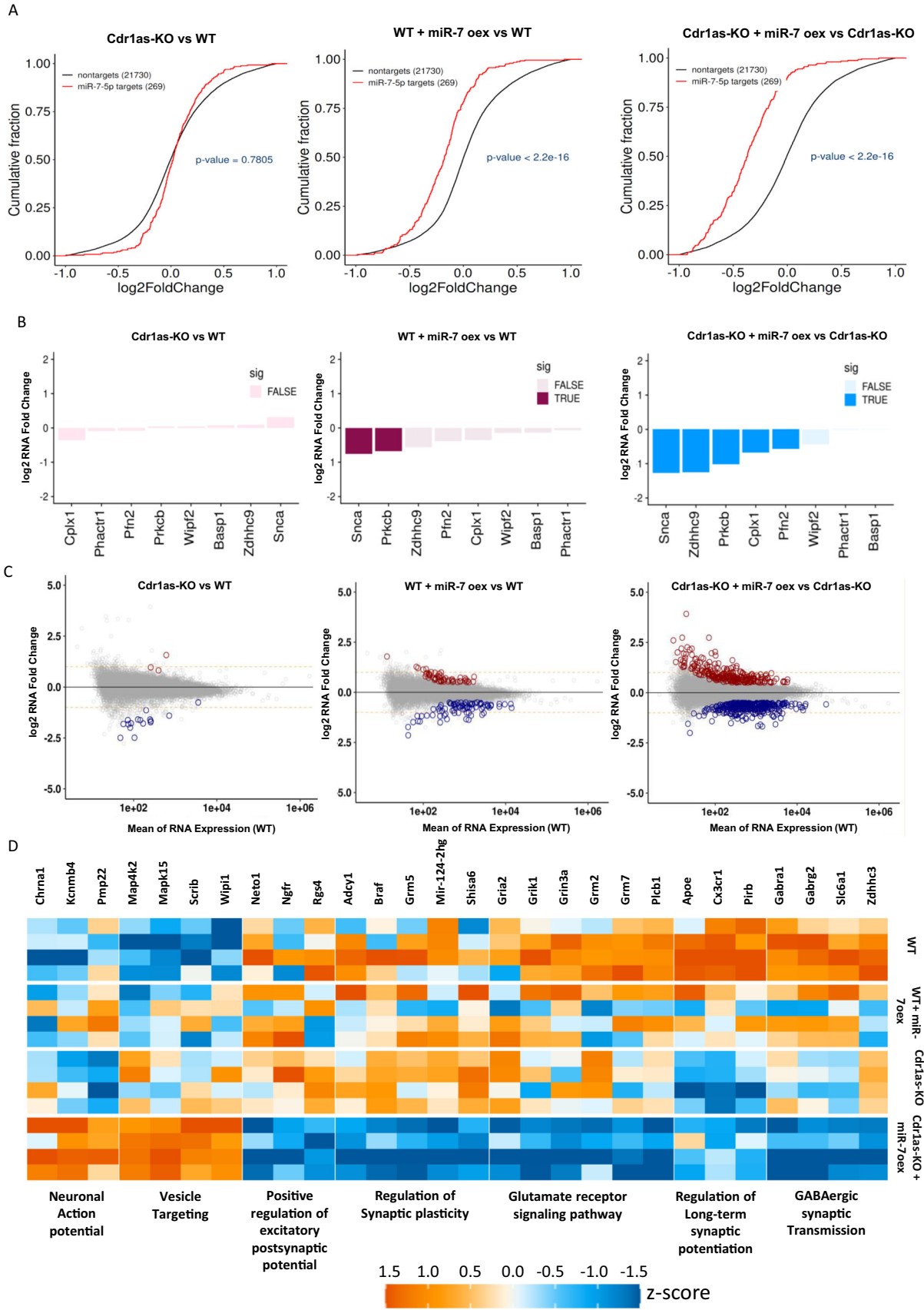

**Figure 5. Transcriptomic changes caused by sustained miR-7 overexpression (oex) are enhanced by the loss of Cdr1as and reveal gene regulatory pathways controlled by their interaction.**

(A) Cumulative distribution function (CDF) plot of gene expression comparing all computationally predicted miR-7 targets ("Methods", red) to mRNAs lacking predicted miR-7 target sites (non-targets, black), across four independent biological replicates of WT, Cdr1as-KO, WT + miR-7 overexpression, Cdr1as-KO + miR-7 overexpression. $P$ value: $U$ Mann–Whitney test. (B) miR-7 target genes involved in Insulin granules secretion, experimentally validated by Latreille et al, 2014 in pancreatic β-cells. Bar plots, the mean log2-fold change of four independent biological replicates per condition, for each comparison tested. Transparent bars, not significant changes (FDR > 0.05). (C) mRNA expression changes for each comparison tested (Cdr1as-KO vs WT, WTmiR-7 oex. vs WT, Cdr1as-KO miR-7 oex.vs Cdr1as-KO). Plotted is the mean change of 4 independent biological replicates per condition. Red dots: statistically significant upregulated genes. Blue dots: statistically significant downregulated genes ("Methods"). (D) Heatmap of statistically significant DE genes associated with GO terms significantly enriched specifically in Cdr1as-KO + miR-7 overexpression vs Cdr1as-KO comparison (specifically up- and downregulated). Each row represents one independent biological replicate. Z-scores across samples were calculated based on normalized and batch-corrected expression values. Source data are available online for this figure.

For example, 24.4% of all upregulated genes (132) are exclusively changing in Cdr1as-KO neurons after miR-7 overexpression (Fig. 5C, right panel; Appendix Fig. S5A: upper diagram, gray ellipse).

This group of genes does not belong to predicted miR-7 target mRNAs and their gene ontology enrichment analysis linked them with vesicle trafficking and membrane-related proteins, and with membrane receptor pathways (Appendix Fig. S5B–D), Similarly, 23.3% of all downregulated genes (151) are only perturbed in Cdr1as-KO and their most overrepresented ontology term associates them with plasma membrane proteins with catalytic activity (Appendix Fig. S5B–D). On the contrary, only a small number of genes are regulated by miR-7 specifically in WT neurons (3.9% upregulated − 0.6% downregulated; Appendix Fig. S5A, yellow ellipse).

In summary, our data suggests that global transcriptomic changes caused by miR-7 overexpression are enhanced by the constitutive loss of Cdr1as. Therefore, we next turned into a more detailed analysis of specific molecular pathways that might explain these differences and additionally connect "non-miR-7- target" gene changes with the rescue of local and network neuronal activity and of glutamate secretion.

### Identification of gene regulatory pathways controlled by the interaction of miR-7 and Cdr1as

We performed gene ontology (GO) enrichment analysis with all differentially expressed genes on our transcriptomic data. We identified statistically significant enriched or depleted GO terms in each of the four conditions tested (WT, Cdr1as-KO, WT + miR-7 oex and Cdr1as-KO + miR-7 oex; Fig. 5D; Appendix Fig. S5B–D; Dataset EV2). Analyses of the GO terms of genes exclusively enriched or depleted after sustained miR-7 overexpression in Cdr1as-KO (Appendix Fig. S5A, upper and lower diagram, red ellipse: upregulated= 44.4% and downregulated = 35.2% of differentially expressed genes) revealed that (1) the largest and specifically upregulated set of genes are related to the control of neuronal action potentials and synaptic vesicles targeting to destination membranes; (2) excitatory postsynaptic potentials, regulation of plasticity and long-term potentiation (LTP), together with regulation of GABAergic transmission and metabotropic and ionotropic glutamatergic receptors signaling pathways are specifically downregulated in Cdr1as-KO neurons after miR-7 overexpression (Fig. 5D; Appendix Fig. S5D; Dataset EV4).

Strikingly, all these GO terms enriched or depleted in WT neurons were conversely depleted or enriched in Cdr1as-KO neurons with miR-7 overexpression, while WT with miR-7 overexpression and Cdr1as-KO control represent intermediates between these two scenarios (Fig. 5D). These finding directly associate what we observed for these same conditions on neuronal activity (Fig. 3).

Altogether, transcriptomic changes validated that the synaptic phenotypes we observed in Cdr1as-KO neurons after sustained miR-7 upregulation are, in part, explained by the interaction between miR-7 and Cdr1as. This interaction modulates gene pathways related to excitatory/inhibitory imbalance and synaptic plasticity. Indeed, we found specific strong downregulation of GluR, GluK1, mGluR, GluN genes, together with GAT1 and GABA-AR, the latest a direct miR-7 target (Fig. 5D). All of them consistently relate molecular changes to the defects in excitatory transmission, local, and network activity.

## Discussion

The secretory role of miR-7 in the pancreas (LaPierre and Stoffel, 2017; Latreille et al, 2014), the evolutionary close relationship between neuronal and pancreatic differentiation lineages (Zhao et al, 2007), as well as the fact that miR-7 evolved within the bilaterian neurosecretory brain (Christodoulou et al, 2010), led us to hypothesize that, in mammalian cortical neurons, miR-7 is also involved in regulating secretion. However, as our data show, in primary cortical neurons, miR-7 is expressed at low levels (~40 miR-7 molecules per neuron in resting state; Fig. EV4D), while Cdr1as is very highly expressed (~250 molecules/neuron, Fig. EV2A). Intriguingly, this is the exact converse in murine pancreatic cells where Cdr1as expression is extremely low while miR-7 is expressed highly (Xu et al, 2015).

How can one explain these expression patterns? Secretion of synaptic vesicles (neurotransmitters) in neurons must be fast and adaptive to stimuli. We thus speculated that Cdr1as has evolved in the mammalian brain to function as a cellular surveillance system to control, depending on external stimuli, the action of miR-7 on its mRNA targets involved in regulating glutamate secretion.

Indeed, in this study, we demonstrated that cortical neuronal adaptation to 15 min of neuronal stimulation causes a rapid transcriptional upregulation of the circular RNA Cdr1as and a post-transcriptional stabilization of miR-7. Even more, the expression dynamics of Cdr1as, Cyrano and miR-7 appear tightly regulated depending on the temporality of neuronal depolarization. While Cyrano appears to be the principal regulator of miR-7 levels upon short depolarization (5 min), 15 min of depolarization

## A  Resting Neurons

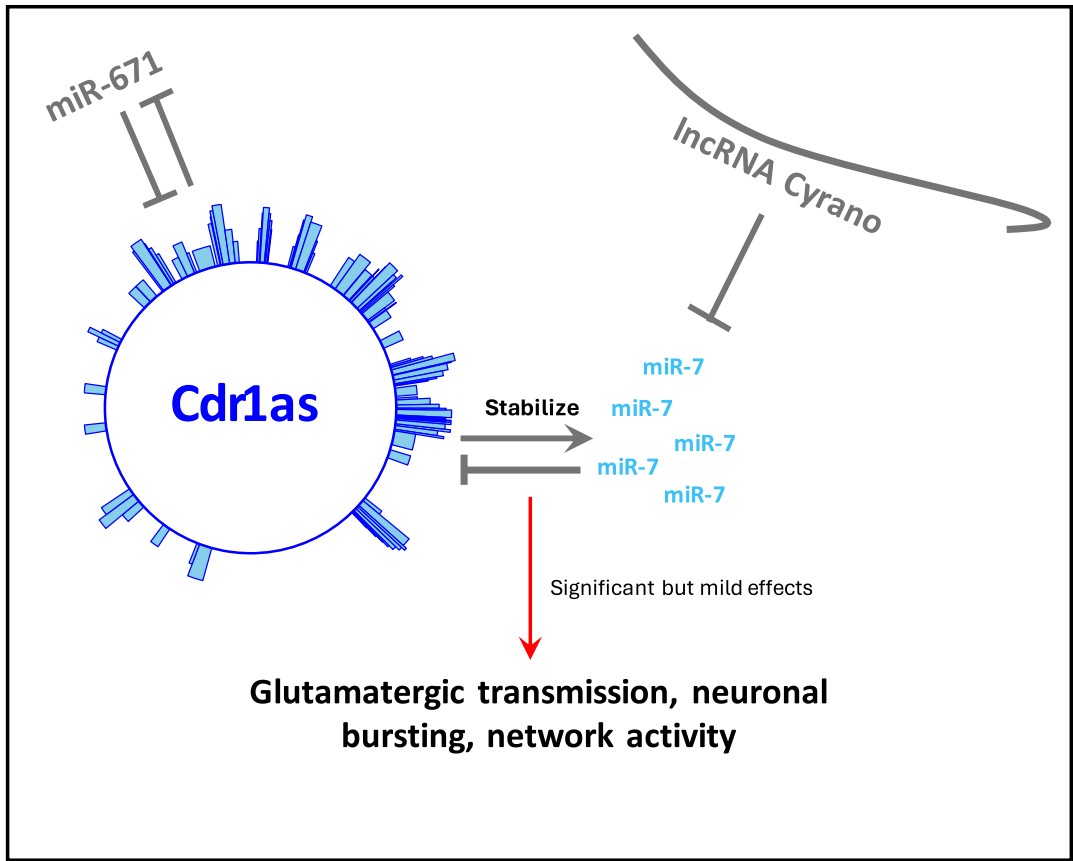

## B  Stimulated Neurons

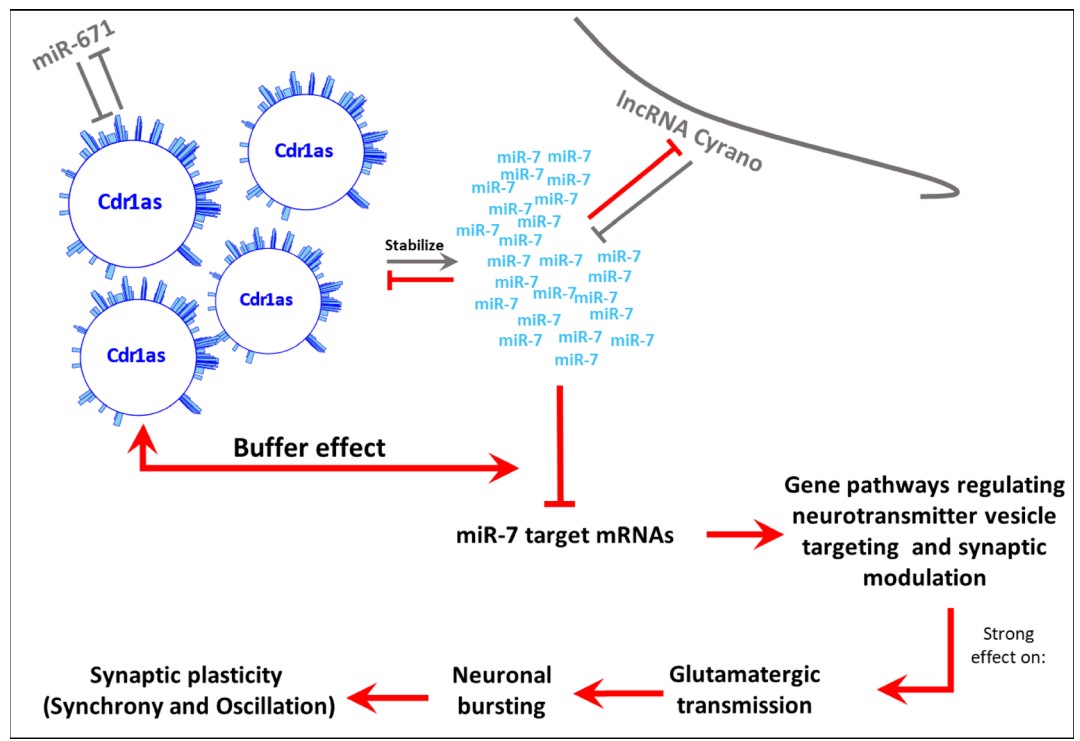

Figure 6.  **Proposed coordinated mechanism and function of Cdr1as and miR-7 in resting and stimulated neurons.**

Gray arrows: Published data (Hansen et al, 2011; Memczak et al, 2013; Hansen et al, 2013; Piwecka et al, 2017; Kleaveland et al, 2018). Red Arrows: New interactions propose in this study. (A) Binding sites of miR-7 to Cdr1as (blue bars) are drawn following Piwecka et al 2017. In resting neurons upon removal of Cdr1as, mir-7 levels are reduced and there are mild but significant effects on glutamate presynaptic release, which induce changes in synchrony and oscillation of the neuronal network. However, the low expression of miR-7 relative to the large amount of Cdr1as impede the continuous action of mir-7 on its targets. (B) Strong sustained neuronal stimulation transcriptionally upregulates Cdr1as which prevents miR-7 degradation and therefore rapidly post-transcriptionally stabilizes miR-7. This persistent increase in miR-7 will cause long-lasting downregulation of Cdr1as from neurites, so that the buffer can self-regulate and therefore dynamically modulate glutamatergic transmission and neuronal network activity. We propose that the post-transcriptional nature of this buffer system allows fast and local regulation of vesicle targeting in synapses.

activate Cdr1as and from then on int takes over miR-7 stabilization. Probably as buffer for miR-7 availability on specific RNA targets. Both Cdr1as and miR-7 are upregulated until 1 h of depolarization and is in cases comparable to canonical immediate early genes under the same conditions) (Fig. 1B–D)). In addition, we discovered that activation of excitatory synaptic transmission (bicuculline) is enough to significantly upregulate Cdr1as and Cyrano, but that the counteracting roles of these two molecules (stabilization—TDMD) maintain total levels of miR-7 stable (Fig. 1G).

Together both stimulation paradigms showed us that: (1) there is a direct and long-lasting transcriptional control over the levels of Cdr1as expression, as stabilizer of miR-7, where Cyrano is not affected. (2) There might be a complementary synaptic mechanism of RNA regulation that keeps levels of miR-7 stable by modulating Cdr1as and Cyrano.

Moreover, our data show that presynaptic terminals of Cdr1as-KO cortical neurons have increased release of glutamate only in a small group of neurons in resting conditions and massively after strong stimulation trains (Figs. 2A,B and  EV3A,B).

The increased secretion of excitatory neurotransmitter is completely reverted by miR-7 upregulation (Fig. 2F). Surprisingly when same Cdr1as-KO neurons are synaptically stimulated by bicuculline, we do not observe synaptic adaptation (Fig. 2C), even though the neurons are chronically active due to higher glutamate release. This suggests compensatory mechanisms might be in place to keep the overall network stable. A similar phenotype has been reported for hippocampal and cortical neurons exposed to chronic excitation treatments, where ultimately glutamate fails to induce LTP. This mechanism is proposed to be a robust type of neuronal protection in excitable environments, like tolerance to ischemia, epilepsy, among others (Tauskela et al, 2008; Ueda et al, 2022). The proposed cellular mechanism responsible of regulation of chronic activation is dependent on NMDA glutamate receptors, which decreases Ca2+influx in the chronically excited neurons. This indicates that stressful requirement could be minimized by the stimulation of primarily synaptic NMDA receptors (Ueda et al, 2022). Interestingly, our RNA-seq data show that the main number of genes downregulated in Cdr1as-KO neurons after miR-7 overexpression are glutamatergic receptors and among them ionotropic NMDA and Kainate receptors: Grik1, Grin3a, Grin2a; the last being a direct target of miR-7 (Fig. 5D; Dataset EV3).

We also show that in a resting state changes in glutamate release have functional consequences on real-time synaptic activity properties, both at local and at network levels (Fig. 3). Cortical neurons with a constitutive loss of the Cdr1as locus, and a significant downregulation of miR-7 (Fig. EV4B), showed upregulated APs frequencies (Fig. 3A), which we could directly associate with the increased release of glutamate from the presynaptic

terminals (Fig. 2). This accelerated AP activity then allows longer and more frequent neuronal bursting, increasing over time of maturation (Fig. 3B,C). This is more strikingly reflected in a globally uncoordinated neuronal network connectivity (Fig. 3D–F), which could potentially alter long-term plasticity responses. All neuronal network phenotypes induced by the lack of Cdr1as are reverted by the sustained overexpression of miR-7, similar to what we observed for glutamate release.

Sustained miR-7 overexpression upregulated equally the number of miR-7 molecules in WT and Cdr1as-KO neurons (Figs. 2E and EV4C). In this context, miR-7 was sufficient to recover any synaptic dysregulation measured in Cdr1as-KO neurons. Meanwhile synaptic activity of WT neurons was only mildly or not affected, depending on the plasticity parameter tested (Fig. 3D–F). Furthermore, we observed that this is also true for miR-7-dependent target gene modulations and global transcriptomic changes, which heavily depend on the expression levels of Cdr1as (Fig. 5). This, together with the observation of depleted Cdr1as molecules in WT neurons after miR-7 overexpression (Fig. 4), proposes that high expression of Cdr1as is critical to control miR-7-dependent effects on bursting regularity, synchrony and oscillation and therefore the balance of excitatory and inhibitory signals.

The clearance of Cdr1as from neurons (Fig. 4) could explain the differences in the strength of glutamate and neuronal activity phenotypes between Cdr1as-KO and WT neurons after sustained miR-7.

This might indicate that the role of Cdr1as as a miR-7 buffer is more relevant in neuronal projections, where a permanent and fast response to sensory stimuli is needed. Therefore, if neurons evolved Cdr1as:miR-7 molecular mechanism of surveillance that can rapidly up- or down-regulate mRNAs locally in synaptic vicinities, this would shape the local modifications on the corresponding synaptic terminals. Some evidence to back up this hypothesis has been presented recently by Zajaczkowski et al (2023), where local dendrite downregulation of Cdr1as, induces an upregulation of miR-7 and impaired fear extinction memory. Therefore, this suggests the participation of circRNA activity localized at the synapse in the process of memory formation (Zajaczkowski et al, 2023).

We propose that one of the potential cellular mechanisms through which miR-7, controlled by Cdr1as, directly regulates long-term changes in neuronal connectivity, can be observed in our measurements of network oscillation (Fig. 3D). In general, oscillation differences account for alternating periods of high and low activity of neuronal communication and changes in network periodicity, which is a hallmark of functional neuronal networks with mixed cell populations of excitatory and inhibitory neurons, such as in cortex, composed of glutamatergic and GABAergic cells. Network oscillation augmentation in Cdr1as-KO, which are then rescued by the sustained upregulation of miR-7, are an indication of an imbalance between glutamate and GABA signaling affecting Cdr1as-KO cortical network activity. In fact, this is corroborated by the glutamate secretion assay,

where we see the increase excitatory signaling in Cdr1as-KO being rescued by miR-7 expression itself (Fig. 2F). Based on the enhanced glutamate presynaptic release observed in Cdr1as-KO (Fig. 2A,B), we expected that miR-7 overexpression and the complete absence of Cdr1as should thus have opposite effects on secretion genes. This is the case for *Snca*, which is one of the strongest regulated miR-7 target genes. In cortical neurons, activity-dependent presynaptic glutamate release depends positively on the concentration of α-synuclein (Sarafian et al, 2017).

Transcriptome analysis allowed us to get a global view of gene expression changes upon miR-7 overexpression and find direct connection between the functional assays (glutamate release, synaptic modulation and electrophysiological defects) and miR-7 molecular signatures. We considered not only the predicted miR-7 targets, but also the mRNA targets that have been experimentally verified (Fig. 5B; Latreille et al, 2014). As a result, we were able to demonstrate that the miR-7 targets identified in our research as regulated by both Cdr1as and miR-7 align with genes previously confirmed as direct miR-7 targets and essential elements of the exocytotic machinery (LaPierre et al, 2022; Latreille et al, 2014). These targets have already been confirmed as miR-7 targets through firefly luciferase reporter assays, which include *Snca, Cspa, Cplx1, Pkcb, Pfn2*, and *Zdhhc9* 3′-UTR reporters (Latreille et al, 2014). The validation of this group of targets, whose expression is common in both secretory glands and cortical neurons, strongly suggests that miR-7 plays a role in regulating genes associated with exocytosis and secretion in cortical neurons.

An interesting example of a Cdr1as-KO-specific downregulated direct miR-7 target gene is *Cplx1* (Fig. 5B). *Cplx1* gene codifies for a synaptic protein that positively regulates a late step in exocytosis of various cytoplasmic vesicles (Kümmel et al, 2011). Most remarkably, it has been reported that the generation of a brain-specific conditional Cplx1-KO mouse results in the attenuation of spontaneous synchronous synaptic function without affecting vesicle priming (López-Murcia et al, 2019). Our data showed a similar hyper-synchronicity phenotype in Cdr1as-KO neurons when miR-7 is upregulated (Fig. 3D) as the one showed by Cplx1-KO animals.

Nevertheless, the functional phenotypes observed in our cortical system are more likely not due to the dysregulation of a single gene, but rather the sum of all gene expression changes generated by miR-7– Cdr1as interaction in complex interconnected pathways. Which, may result in the observed defects. To account for the complexity of the miR-7 effects, we performed a gene regulatory network analysis based on a machine learning algorithm trained with all differentially express genes in each genotype and condition. This analysis pointed out the most regulated and interconnected miR-7 target genes, potential key gene regulatory players connecting gene ontology terms with our functional phenotypes (Appendix Extended Discussion; Appendix Figs. S6 and S7).

The exact involvement of other network partners such as lncRNA Cyrano remains to be studied. We did not observe long-term statistically significant change in Cyrano after sustained depolarization (Fig. 1B), which indicates that during neuronal activation the regulation of miR-7 levels are not predominantly regulated by Cyrano. Nevertheless, we did observe upregulation after excitatory synaptic stimulation (Fig. 1F).

Therefore, we can speculate that Cdr1as is the regulator dominating miR-7 cellular availability, due to its prevailing

stabilization of miR-7, but Cyrano could have a local increasing due to specific glutamatergic activation. Other less explored players, like Hu RNA binding proteins, which regulate miR-7 processing in non-neural cells through direct binding to the pri-miR-7 could also be involved (Lebedeva et al, 2011). On the other hand, we did not observe a general downregulation of Cyrano after sustained overexpression of miR-7 in WT neurons, but only in Cdr1as-KO neurons (Fig. 4F). Consequently, our data confirmed a direct regulation of miR-7 over Cyrano expression, but this might not be as strong in cortical neurons as in cerebellar neurons, as previously shown by Kleaveland et al (2018). To understand better the dynamics of miR-7 regulation, it will be interesting to test perturbations on Cdr1as, miR-7, and Cyrano during neuronal stimulations.

Altogether, our main hypothesis is that a potential explanation of the differentially enhanced effect of miR-7 activity in Cdr1as-KO neurons is that Cdr1as act as the coordinator of miR-7 activity, triggered by neuronal stimulation. We demonstrate that upon neuronal depolarization, miR-7 is post-transcriptionally stabilized (Fig. 1). Therefore, the proxy of miR-7 overexpression would mimic the long-term accumulation of miR-7 generated by phenomena such as a long-standing increase in glutamate release or other similar log-term synaptic potentiation, a common feature observed in many major neuropsychiatric disorders, such as depressive disorder and autistic spectrum syndrome (Kasper and McEwen, 2008; Gutierrez et al, 2009). Subsequently, as Cdr1as buffers the direct action of miR-7 on its targets, the absence of the circRNA favors direct interaction of miR-7 with its respective targets, which subsequently has other indirect transcriptomics effects as well (Fig. 5).

Finally, we propose a new molecular model (Fig. 6) for the action mechanism by which Cdr1as and miR-7 exert their function.

In resting neurons (Fig. 6A) the removal of Cdr1as causes mild reduction of miR-7 levels and affects glutamate presynaptic release, clearly reflected in neuronal network modulations. However, the low basal expression of miR-7 relative to the large amount of Cdr1as, maintain action of miR-7 on secretion targets stable.

During strong sustained neuronal stimulation (Fig. 6B) Cdr1as is transcriptionally upregulated preventing miR-7 degradation and rapidly post-transcriptionally stabilizing it. Nevertheless, short or synaptically local stimulation (bicuculline), briefly upregulates Cyrano as well, which keeps miR-7 buffered and controls its activity. Long-lasting persistent increase of miR-7 expression will downregulate Cdr1as, so that the buffer system can self-regulate and therefore dynamically modulate glutamatergic transmission and neuronal network activity. However, constitutive removal of Cdr1as prevents network adaptation to synaptic stimuli. The post-transcriptional nature of this buffer system could allow fast and local regulation of vesicle targeting in synapses and ultimately synaptic plasticity.

# Methods

## Animals

### Cdr1as knockout mice

Cdr1as knockout (Cdr1as-KO) strain was generated and maintained on the pure C57BL/6N background (Piwecka et al, 2017). All

animals used in this work came from intercrosses of hemizygous Cdr1as-/Y males and heterozygous Cdr1as + /− females in (F3 generation from original founder), and mutants were compared to wild-type littermate controls. Animals were kept in a pathogen-free facility in a 12 h light–dark cycle with ad libitum food and water. Animal care and mouse work were conducted according to the guidelines of the Institutional Animal Care and Use Committee of the Max Delbrück Center for Molecular Medicine, the Landesamt für Gesundheit und Soziales of the federal state of Berlin (Berlin, Germany). Genotyping of animals was conducted as previously described (Piwecka et al, 2017).

## Biochemical and molecular biology

### Cloning, transformation, and propagation of plasmids

Cloning was performed according to standard methods. Plasmids were cut using appropriate restriction enzymes (Fast Digest - Thermo Scientific), dephosphorylated with FastAP and purified from an agarose gel with a Zymoclean Gel DNA Recovery kit (Zymo Research). Inserts were either PCR-amplified, then digested with the same set of restriction enzymes or oligonucleotides with fitting overhangs were annealed and phosphorylated with T4 PNK. Insert and cut plasmid were ligated with T4 DNA Ligase and subsequently transformed into chemically competent bacteria (Mix&Go Competent Cells, DH5α) in the presence of the corresponding antibiotic for selection. Finally, plasmids were isolated using ZymoPURE Plasmid Miniprep or ZymoPURE Plasmid Midiprep kit from bacterial culture. For plasmids obtained from Addgene, the same procedure of propagation and plasmid purification was used (Addgene Plasmid #99126 and #114472).

### Genotyping of mutant animals

To genotype Cdr1as-KO strain, genomic DNA was extracted from tail cuts taken from newborn or adult animals. Tissue was digested using QuickExtract™ DNA Extraction Solution 1.0 at 65 °C for up to 30 min depending on the tissue size, reaction was stopped at 98 °C for 2 min and 1 μl of the supernatant, containing the genomic DNA, was used for the follow up end-point PCR detection of wild-type or mutant splice sites. Genotypes were determined by end-point PCR performed on corresponding template genomic DNA using a set of primers for 3'SS multiplex detection (3'SS_For + 3'SS_Rev + 5'SS+For). PCR products for wild-type (355 bp), heterozygotes (418 and 355 bp), and knockout (418 bp) animals were analyzed on a 2.5% agarose gel.

### Nanostring

To achieve direct quantification of targeted single RNA molecules in a sample of total RNA, 100 ng of total RNA extract was incubated with a 72-plex Core Tag set, custom-made probe mix (reporter + capture probes) and corresponding hybridization buffer at 67 °C for 18 h, followed by a cooling down step at 4 °C for 10 min. All samples were diluted to a final volume of 30 μl and loaded on the nCounter gene expression cartridge (12 sample panel). Quantification of RNA molecules was done by nCounter SPRINT™ Profiler instrument (Nanostring Technologies, USA) according with manufacturer instructions. Normalized counts of RNA molecules were obtained using nSolver™ Data analysis software. Custom probes panel in Dataset EV6.

### miR-7 overexpression

The sequence of pri-miR-7a was amplified from mouse brain tissue and cloned into an AAV vector (Addgene Plasmid #99126), driven by a neuronal promoter (hSyn1) and expressed together with mCherry fluorescent reporter. The final viral particles (AAV serotype 9 and mutated versions of Serotype 9) were generated in Charité Viral Core Facility. AAV that only expressed mCherry protein under the same neuronal promoter was used as control condition (Addgene Plasmid #114472). Wild-type and Cdr1as-KO neurons were infected at DIV 4-6 by directly adding the viral particles into the culturing media at a titer of $10^9$ VG/ml. Half of the media was exchanged 5 days post-infection and then once a week. At DIV21, cells were fixed or harvested in Trizol dependent on the downstream experiments.

### Quantitative real-time PCR (qPCR)

Quantitative real-time PCR experiments were conducted on an ABI StepOne Plus instrument or Roche LightCycler 96 System. cDNA was diluted up to 20X with ddH2O and mixed with 2× Biozym Blue S'Green + ROX qPCR Master Mix and 1 μl of 10 μM Primer mix to obtain 20 μl of total volume per well. All measurements were conducted at least in three technical replicates. For miRNA expression analysis, cDNA was diluted 3× with ddH2O, and TaqMan assays (Applied Biosystems) were used along with TaqMan Universal master Mix II, no UNG-1 (Applied Biosystems). Relative quantification of gene expression was performed by using the comparative ΔΔCT method (Pfaffl, 2001). Primer list in Dataset EV5.

### Reverse transcription

Total RNA was reverse transcribed to cDNA using Maxima H Minus Reverse Transcriptase (Thermo Scientific) (100 U/μg RNA). First, 0.5–1 μg of total RNA was mixed with 1 mM dNTPs, 0.5 μg/μl random hexamers, denatured at 65 °C for 5 min and then placed on ice. Immediately, 200 U/μl of Reverse Transcriptase, 4 μl 5× RT Buffer, and 0.5 μl RiboLock (40U/μl) were added. The cDNA synthesis was carried out as follows: 10 min at 25 °C, 1 h at 50 °C, 5 min at 85 °C. For miRNAs, instead of normal random hexamers, gene-specific stem-loop primers (TaqMan RT primers) were used, which only need a short overlap with the miRNAs 3'end. First, 100 ng of total RNA was reverse transcribed using SuperScript III (Invitrogen) (100 U/reaction), 1 mM dNTPs, 20 U Ribolock (40 U/ μl), 1× TaqMan RT primer per each gene of interest (miR-7a, miR-671, Let-7a, snoRNA202, U6 snRNA) (Applied Biosystems) and 1× first-strand synthesis buffer were filled up to 20 μl reaction volume with ddH2O and mixed. Then, the gene-specific cDNA synthesis was carried out as follows: 30 min at 16 °C, 30 min at 42 °C, and terminated for 5 min at 85 °C.

### RNA extraction

Total RNA from primary neurons was extracted using a homemade TRIzol reagent in combination with Direct-zol RNA Kit (Zymo Research). The samples were fixed in cold methanol for 10 min at −20 °C, then washed in cold PBS, TRIzol was directly added to each well and cells were scraped away to ensure full detachment of all neuronal processes. The dissociated sample was collected in a tube with silica beads and homogenized at 5500 rpm 2× for 20 s. Then equal volume of 100% Ethanol was added directly to the sample and then transferred to a Zymo-Spin™ IIICG Column and

centrifuged (30 s × 16,000 × g). The flowthrough was discarded, and DNAse I (Zymo Research) treatment followed, incubating the column with 6U/μl for 15 min at room temperature. Afterwards, three consecutive washing steps were performed and finally the RNA was eluted from the Column in DNAse/RNAse-free water.

## Imaging techniques and analysis

### smRNA FISH (Stellaris) (Raj et al, 2008)

Single-molecule fluorescent in situ hybridization (smFISH) protocol was performed in wild-type and Cdr1as-KO primary neurons which were fixed at 14–21 DIV. Stellaris oligonucleotide probes complementary to Cdr1as, Cyrano (Oip5-as1), Hprt, circHipk3 or linHipk3 were designed using the Stellaris Probe Designer (LGC Biosearch Technologies) as conjugates coupled to Quasar 670 or Quasar 570. Neurons were fixed in 4% Formaldehyde for 10 min. and probes were hybridized overnight in 10% formamide at 37 °C and 100 nM final concentration. The next day, the coverslips with neurons were washed according to Stellaris protocol (Biosearch Technologies), incubated with DAPI 5 ng/ml in the second wash and mounted with Gloxy buffer. Images were acquired on an inverted Nikon Ti-E microscope with 60× oil NA1.4 objective and Andor iXON Ultra DU-888 camera; Z stacks had 0.3-μm spacing and were merged by maximum intensity projection. Dot detection was performed using StarSearch software from Raj Lab (https://github.com/arjunrajlaboratory/rajlabimagetools/wiki) on manually segmented neurons (somas and neurites). The masks for each segmented neuron were created manually by utilizing the bright-field channel in the StarSearch software. Initially, the soma was delineated, followed by the definition of at least three neurites per cell. Each neurite had the same length, measuring 300 pixels from the marked soma.

Posterior quantification analysis was done by counting the number of dots (Cdr1as or Cyrano molecules, respectively) and normalized to the area and per compartment. All statistics were calculated using the Mann–Whitney test, nonparametric rank comparisons, between different conditions (GraphPad Prism 9.1.0); statistical significance was assigned for $P < 0.05$. Probe sequences in Dataset EV5.

### smRNA FISH + immunofluorescence (Stellaris+IF) (adapted from Singer Lab)

Single-molecule fluorescent in situ hybridization coupled to immunofluorescence was performed in WT neurons DIV21. Cells were permeabilized/blocked with 0.1% Triton X-100 and 0.5% UltraPure BSA solution, then pre-hybridize in 10% (vol/vol) formamide, 2× SSC, and 0.5% UltraPure BSA. Hybridization of RNA probe together with correspondent primary antibody was carried out overnight in 10% formamide buffer at 37 °C. The next day, cells were washed and secondary antibody was incubated for 30 min in 10% formamide buffer. Images were acquired on an inverted Nikon Ti-E microscope with 60× oil NA1.4 objective and Andor iXON Ultra DU-888 camera; Z stacks had 0.3 μm spacing and were merged by maximum intensity projection.

### Protein–RNA colocalization analyses from Stellaris + IF images

To assess the colocalization of FISH molecules with a specific protein, we measured the distance from each Cdr1as molecule's center to its nearest protein of interest. First, we generated a binary image of the protein channel. This image highlighted pixels where the protein was tagged, utilizing Otsu's thresholding method (Otsu, 1979). Our choice of Otsu's method was informed by its ability to adequately capture protein localization upon manual inspection. This binary image was subsequently transformed into a distance map using Scipy's (Virtanen et al, 2020) Euclidean Distance Transform (EDT). In this transformed image, the value at each pixel indicates its distance to the nearest protein pixel, with protein pixels having a value of zero. Leveraging this distance map, we ascertained the distance from each Cdr1as molecule center (identified via the RS-FISH (Bahry et al, 2022) plugin in FIJI (Schindelin et al, 2012)) to the closest protein pixel. Given that RS-FISH has subpixel precision, we employed bilinear interpolation (weighted sum of the nearest pixels) to account for 1, 2, or 4 pixels in the distance map, depending on whether the coordinates were integers or floats. For every protein under investigation, distances derived from all images were computed and plotted in a histogram.

### Single-molecule miRNA FISH (ViewRNA Cell Plus)

Single-molecule miRNA fluorescent in situ hybridization (miRNA smFISH) protocol was performed in wild-type and Cdr1as-KO primary neurons at DIV21. miRNA probes for mature miR-7a-5p were obtained commercially (Thermo Scientific) and labeled with A670. miRNA smFISH protocol was performed according to manufacturer protocol (ViewRNA Cell Plus—ThermoFisher) with the modification, i.e., with the addition of protease K (Invitrogen) treatment in a dilution 1/1000 for 10 min. before the miRNA probe hybridization step. The coverslips were mounted with Prolong Gold Antifade mounting media (Invitrogen) and imaged using Keyence BZ 9000 or Leica Sp8 confocal microscope. Image processing was done using Fiji-ImageJ.

### Colocalization analyses of RNA-RNA in smFISH

Colocalization of circRNA Cdr1as and lncRNA Cyrano molecules in neurons was measured based on MNN distances and probability of random association calculation. smRNA FISH (Stellaris) images of Cdr1as and Cyrano were used as the test condition. Cdr1as and Tfrc mRNA (transferrin receptor C mRNA) as negative control. Images of two probes within the same RNA molecule (linHipk3 and circHipk3) as a positive control of true colocalization. Dot detection was performed using StarSearch software from Raj Lab (https://github.com/arjunrajlaboratory/rajlabimagetools/wiki) on manually segmented neurons (somas and neurites). The analyses to measure Intermolecular Distances and Determining the Significance of association, were done based on the previously published work (Eliscovich et al, 2017), but rewritten in a new R script according to our requirements.

### Synaptic glutamate release

i.  iGluSnFR

AAV1 particles containing a Glutamate sensor under the human Synapsin-1 promoter (pAAV.hSyn.iGluSnFr.WPRE.SV40 (Borghuis et al, 2013) were obtained from A.W. Lab at MDC Berlin. Primary neuronal cultures were infected at DIV 4-6 with the AAV particles to express the glutamate sensor at the cell surface and then recorded at DIV17-21.

ii. Image acquisition

All image acquisition was done as previously described (Farsi et al, 2021). In brief, neurons were incubated at room temperature in Tyrode's buffer (120 mM NaCl, 2.5 mM KCl, 10 mM glucose, 10 mM HEPES, pH 7.4, osmolality was adjusted to that of the culture medium with sucrose). Action potential-evoked glutamate release was recorded in a chamber with two electrodes for electrical stimulation, mounted on a Nikon Eclipse Ti inverted microscope equipped with a PFS focus controller, a prime 95B scientific CMOS (Photometrics) camera and a pulse generator (HSE- HA, Harvard Apparatus). Emission was collected with a 60×, 1.49 NA Nikon objective. Evoked glutamate release was performed by continuous imaging at 20 Hz after application of 20 field stimuli at 0.5 Hz in the presence of 50 μM AP5 (l-2-amino-5-phosphonovaleric acid) and 10 μM CNQX, to block postsynaptic, imaging solution was supplemented with 2 mM $CaCl_2$ and 2 mM $MgCl_2$. Spontaneous events were then captured by 5-min continuous imaging at 20 Hz at 4 mM $CaCl_2$, in the presence of 0.5 μM TTX to block action potential firing.

iii. Image analysis

All image analysis was done as previously described in Farsi et al 2021. In brief, time-lapse images were loaded in MATLAB (Mathworks, Natick, MA) and after de-noising and filtering, active synapses were detected as local maxima on the first derivative over time of the image stack. Regions of interest (ROIs) were defined by stretching the maxima to a radius of 4 pixels (700 nm). Taking the first derivative over time allowed to resolve fluorescence changes associated with evoked or spontaneous release and localize release sites.

iv. Release probability calculation

As previously described in Farsi et al 2021, to characterize the release properties of individual synapses, filtered fluorescence traces were used for peak detection. Peaks with the amplitude seven times greater than standard deviation of baseline trace during spontaneous imaging were counted as a successful glutamate release event. Spontaneous frequency was calculated from the total number of events detected during 5-min acquisition. Evoked probability for each synapse was obtained by dividing the number of events happening within one frame after stimulation by the total number of stimulations.

v. Kernel-Density estimation

All release probability calculations were plotted using the Kernel-Density-Estimate and creating a curve of the distribution of individual measurements. The curve is calculated by weighing the distance of all the points in each specific location along the distribution. Each data point is replaced with a weighting function to estimate the probability density function. The resulting probability density function is a summation of every kernel.

## Cell culture methods

### Primary neuronal culture

Primary cortical neurons were prepared from C57BL/6N mouse pups (P0) as previously described (Kaech and Banker, 2006). In brief, cortices were isolated from male P0 C57BL/6N mice from wildtype and Cdr1as-KO genotypes blindly. The tissue was dissociated in papain at 37 °C and then the single cell suspension was seeded in a previously PLL-coated glass coverslips, to finally place them on top of a monolayer of feeder astroglia. The cells were maintained in culture up to 21 days in Neurobasal-A medium (Invitrogen) supplemented with 2% B-27(Gibco), 0.1% PenStrep (10 kU/ml Pen; 10 mg/ml Strep), and 0.5 mM GlutaMAX-I (Gibco) at 37 °C and 5% $CO_2$. Neuronal cultures were grown blind of genotype to minimize the effects of subjective bias when treating and processing samples. The genotype was only assessed afterward when downstream analyses were performed.

### $K^+$ treatments

On DIV13 neurons were incubated overnight with a mix of: 0.5 μM TTX (tetrodotoxin), 100 μM AP5 (l-2-amino-5-phosphonovaleric acid) and 10 μM CNQX, to silence all spontaneous neuronal responses. On DIV14, neuronal basal media was changed for stimulation media (170 mM NaCl, 10 mM HEPES pH 7.4, 1 mM MgCl2, 2 mM $CaCl_2$), containing whether 60 mM KCl, (2) 50 μM DRB (5,6-dichloro-1-β-D-ribofuranosylbenzimidazole) or just (3) NaCl as osmolarity control. Cells (neuronal cultures and astrocyte-feeder layer) were incubated at 37 °C; 5% $CO_2$ for 5, 15, 30, or 60 min 1 h, fixed with cold methanol for 10 min and used immediately for RNA extraction. Cell death was controlled by staining of stimulated neuronal cultures with Trypan blue (1/10 dilution) and subsequent cell counting, untreated cells we used as a negative control

### Bicuculline + 4-Aminopyridin (4-AP) treatment

On DIV14, neuronal basal media was changed for stimulation media containing: 50 μM Bicuculline + 75 μM 4-Aminopyridine (antagonist of GABA-A receptors and $K^+$ Channel blocker, respectively) or stimulation media containing DMSO as vehicle control. Neuronal cultures were incubated at 37 °C; 5% $CO_2$ for 30 min, then removed from stimulation media and washed once with BPS. Neurons were fixed with cold methanol for 10 min and used immediately for RNA extraction. Cell death was controlled by staining of stimulated neuronal cultures with Trypan blue (1/10 dilution) and subsequent cell counting; untreated cells were used as a negative control.

### Glutamate secretion assay

To assess the levels of secreted Glutamate in Wild-type and Cdr1as-KO neurons with or without miR-7a overexpression, we implemented the luciferase-based Glutamate-Glo™ assay. Neurons were plated in 48-well at equal confluency and media from each well was collected at DIV18-21 and saved at −20 °C until ready to perform the assay. In total, 25 μl of media from each sample were transferred into a 96-well plate to perform the final reaction. A negative control of only buffer was included for determining assay background, and a control of no-cells media was used to determine basal levels of Glutamate. Samples were mixed with Glutamate detection Reagent prepared according to manufacturer instructions and the plate was incubated for 60 min at room temperature. Recording of luminescence was done using the plate-reading luminometer (Tecan M200 infinite Pro plate reader) following manufacturer protocol. A stock solution of 10 mM glutamate was used to create the standard curve (100 μM to $1.28 \times 10^{-3}$ μM) and as a positive control. All Glutamate concentrations were estimated based on standard curve linear regression. Analyses were done by comparison of RLU values across conditions and all statistics were calculated using two-way ANOVA, between different conditions, and $P < 0.05$ was considered as statistically significant.

## Multi-electrode array (MEA) recordings

Cells were seeded in 48-well CytoView MEA plates with 16 poly-3,4-ethylendioxythiophen (PEDOT) electrodes per well (Axion Biosystems). Each well was coated with 100 µg/mL poly-D-Lysine and cells were seeded to a density of 100–150 K mixed with 1 µg/mL laminin (FUDR was added to the media one day after seeding). Neurons were grown in supplemented Neurobasal-A media for the first 7 days, then half of the media was changed to BrainPhysTM, supplemented with B-27, 1 day before each recording. Recordings of spontaneous extracellular field potentials in neurons from DIV7 to DIV21 were performed using a Maestro MEA system and AxIS Software with Spontaneous Neural Configuration (Axion Biosystems, Atlanta, GA, USA). Spikes were detected using an adaptive threshold set to six times the standard deviation of the estimated noise. The plate was first allowed to ambient in the Maestro device for 3 min and then 10 min of raw spikes data was acquired for analysis.

In the case of recordings after bicuculline treatments, on DIV21 neurons were incubated for 1 h with 10 µM of Bicuculline or the corresponding vehicle control (DMSO). Ten minutes of raw spikes data were acquired for analysis before and after 1-h incubation and the difference in neuronal spikes was used to calculated the activity changes (Δ).

## MEA data analysis

For MEA data analysis, we used a sampling frequency of 12.5 kHz, and the active electrode selection criteria was 5 spikes/minute. (1) Mean firing rate (MFR): action potentials (APs) frequency per electrode. (2) Burst Frequency: Total number of single-electrode bursts divided by the duration of the analysis, in Hz. (3) Burst Duration: Average time (sec) from the first spike to last spike in a single-electrode burst. (4) Network Asynchrony: Area under the well-wide pooled inter-electrode cross-correlation, according to Halliday et al, 2006. (5) Network Oscillation: Average across network bursts of the Inter Spike Interval Coefficient of Variation (standard deviation/mean of the interspike interval) within network bursts. (6) Burst Peak: peak of the Average Network Burst Histogram divided by the histogram bin size to yield spikes per second (Hz). The statistics was done using two-way ANOVA mixed model with Bonferroni correction for multiple measurements, between different conditions, statistical significance was assigned for $P < 0.05$.

## Bulk RNA sequencing

### Total RNA libraries and sequencing (for circRNA detection)

In all, 1 µg of total RNA from wild-type and Cdr1as-KO neurons was used as starting material for total RNA libraries. Then, ribosomal RNA (rRNA) was depleted with Ribominus Eukaryote Kit v2 or with RNase H approach (Adiconis et al, 2013). Successful ribodepletion was assessed by Bioanalyzer RNA 6000 Pico Chip. RNA was fragmented and subsequently prepared for sequencing using the Illumina Truseq Stranded Total RNA Library Prep kit. Libraries were sequenced on a Nextseq 500, at 1 × 150 nt.

### PolyA+ libraries and sequencing

All samples from wild-type and Cdr1as-KO neurons, control and miR-7 overexpression, were prepared as described in the TruSeq Stranded RNA sample preparation v2 guide.

In brief, 500 ng of RNA was fragmented, reverse transcribed, and adapter-ligated, followed by a pilot qPCR to determine the optimal PCR amplification. Library quality was assessed using TapeStation (DNA1000 kit) and Qubit. Samples were sequenced on an Illumina NextSeq 500 with 1 × 150 bp.

### Small RNA libraries and sequencing

All libraries were prepared based on total RNA obtained from WT and Cdr1as-KO neurons according to Illumina TruSeq Small RNA Sample Prep Kit and sequenced on a NextSeq 500 with 1 × 50 bp.

### RNA-Seq analysis

RNA-seq reads were mapped to the mouse mm10 genome assembly using STAR v2.7.0a (Dobin et al, 2013). Aligned reads were assigned to genes using annotations from Ensembl (Mus_musculus, release 96) and featureCounts v1.6.0 () with the parameter reverse stranded mode (-s = 2). Differential gene expression analysis was done using DESeq2 v1.30.1 (Love et al, 2014), taking batch effects into account when making comparisons between different genotypes. To assess overexpression effects in each genotype, we used nested effect models that respected the pairing of samples from the same animal. The significance threshold for differential expressed genes was set to an adjusted $P$ value of 0.05, The threshold for the log2-fold-change was set to 0.5.

### miR-7-5p target regulation analysis

To test if log2FoldChange distributions of miR-7 targets are significantly different from the other differentially expressed genes, we performed a two-sided Mann–Whitney $U$ test for each comparison. A list of predicted conserved targets of miR-7-5p and miR-122 as control were downloaded from Data ref: TargetScan Mouse, release 7.2 (Agarwal et al, 2015) and Data ref: miRDB (Chen and Wang, 2020). We only used targets that were listed in both databases.

### Gene ontology (GO) analysis

Gene ontology enrichment analysis was done using topGOtable function in the pcaExplorer (Marini et al, 2019). In GO analysis, genes showing average log2-fold change of 0.5 and adjusted $P$ value less than 0.05 were considered significant and all expressed genes were used as background. We used the elim algorithm instead of the classic method to be more conservative and excluded broad GO terms with more than 1000 listed reference genes.

### Gene networks analysis

To investigate how genes related to neurological phenotypes are associated with gene expression regulation by the noncoding RNA miR-7, we predicted gene regulatory networks using random forests (Breiman, 2001). We started from fifteen genes that were found in the list of differentially expressed genes and were known from the literature to be involved in synaptic activity. We found potential regulators of these genes and possible connections to miR-7a-5p target genes by applying the random forest approach in two steps: First, potential regulators for the expression levels of phenotype-related genes were detected from all other differentially expressed genes, determining for each phenotype-related gene the top ten predictive genes and keeping only predictive genes that overlapped between two or more phenotype-related genes. Second, these genes were connected to direct targets of miR-7 extracted from the

TargetScan Mouse and miRDB databases as described above. Again, the top ten predictive genes were selected and the overlapping genes are shown. In each step, leave- one-sample-out cross-validation was used to test the performance of the random forest models in predicting the expression level of a gene for a previously unseen sample. The prediction performance was evaluated by the root mean squared error (RMSE), which has a minimum of zero, indicating perfect predictions, and no upper bound. In addition, we computed differences of the predicted or ground truth values relative to the training examples and computed the cosine similarity of these two difference vectors, which is 1 if predictions and ground truth show consistent patterns and −1 if predictions and ground truth show opposite patterns. The overall prediction performance on left-out test samples was good, suggesting that the models found meaningful gene relationships. For further validation, the proposed connections were compared to protein–protein interactions from StringDB (10090.protein.info.v11.5).

## Data availability

All RNA-sequencing raw data for all analyzed samples have been deposited in GEO: accession GSE224184: Go to https://www.ncbi.nlm.nih.gov/geo/query/acc.cgi?acc=GSE224184. Output data from RNA-seq statistical analyses, GO term enrichment, primers. Source data for the live imaging of presynaptic glutamate release from WT and Cdr1as- KO neurons are available in BioStudies Repository: Go to https://www.ebi.ac.uk/biostudies/studies/S-BSST1417.

The source data of this paper are collected in the following database record: biostudies:S-SCDT-10_1038-S44319-024-00168-9.

## Peer review information

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

## Acknowledgements

The authors thank all past and present members of the Rajewsky lab for helpful discussions and advice. We thank Margareta Herzog for all organizational help. CACJ thanks Christine Kocks and Anastasiya Boltengagen for technical advice on single-molecule FISH, Thomas Müller for help with primary neuronal cultures, Ivano Legnini for advice on neuronal stimulation experiments, Ilan Theurillat for feedback on the manuscript edition and Petra Stallerow for technical support with animal caretaking. SJK thanks Marcel Schilling and Marvin Jens for computational analysis assistance. EG thanks Lisa Barros de Andradre e Sousa and Marie Piraud for the initial discussions. NR thanks DFG Leibniz Award and DFG Neurocure/BrianBank for support. CACJ was funded by the MDC graduate program, DZHK project 81×2100155 and DFG Neurocure/BrianBank. SJK was funded by EU ITN—circular RNA Biology Training Network: circRTrain (721890) and DFG Excellence Cluster (DFG EXC2049). GZ was funded by DFG Leibniz Award. EG was funded by Helmholtz Association's Initiative and Networking Fund through Helmholtz AI. MP has been supported by the Polish National Agency for Academic Exchange (Polish Returns grant no. PPN/PPO/2019/1/00035/U/0001) and the National Science Centre (grant no. 2018/30/E/NZ3/00624).

## Author contributions

**Cledi A Cerda Jara**: Conceptualization; Data curation; Formal analysis; Validation; Investigation; Visualization; Methodology; Writing—original draft; Writing—review and editing. **Seung Joon Kim**: Data curation; Formal analysis; Investigation; Visualization. **Gwendolin Thomas**: Data curation; Investigation; Methodology. **Zohreh Farsi**: Data curation; Formal analysis; Visualization; Methodology. **Grygoriy Zolotarov**: Data curation; Formal analysis; Visualization; Methodology. **Giuliana Dube**: Validation. **Aylina Deter**: Validation. **Ella Bahry**: Software; Formal analysis; Validation; Visualization; Methodology. **Elisabeth Georgii**: Data curation; Software; Formal analysis; Visualization; Methodology; Writing—review and editing. **Andrew Woehler**: Data curation; Formal analysis; Methodology. **Monika Piwecka**: Supervision; Investigation; Methodology; Writing—review and editing. **Nikolaus Rajewsky**: Conceptualization; Supervision; Funding acquisition; Investigation; Writing—original draft; Project administration; Writing—review and editing.

Source data underlying figure panels in this paper may have individual authorship assigned. Where available, figure panel/source data authorship is listed in the following database record: biostudies:S-SCDT-10_1038-S44319-024-00168-9.

## Funding

## Disclosure and competing interests statement

The authors declare no competing interests.

# Expanded View Figures

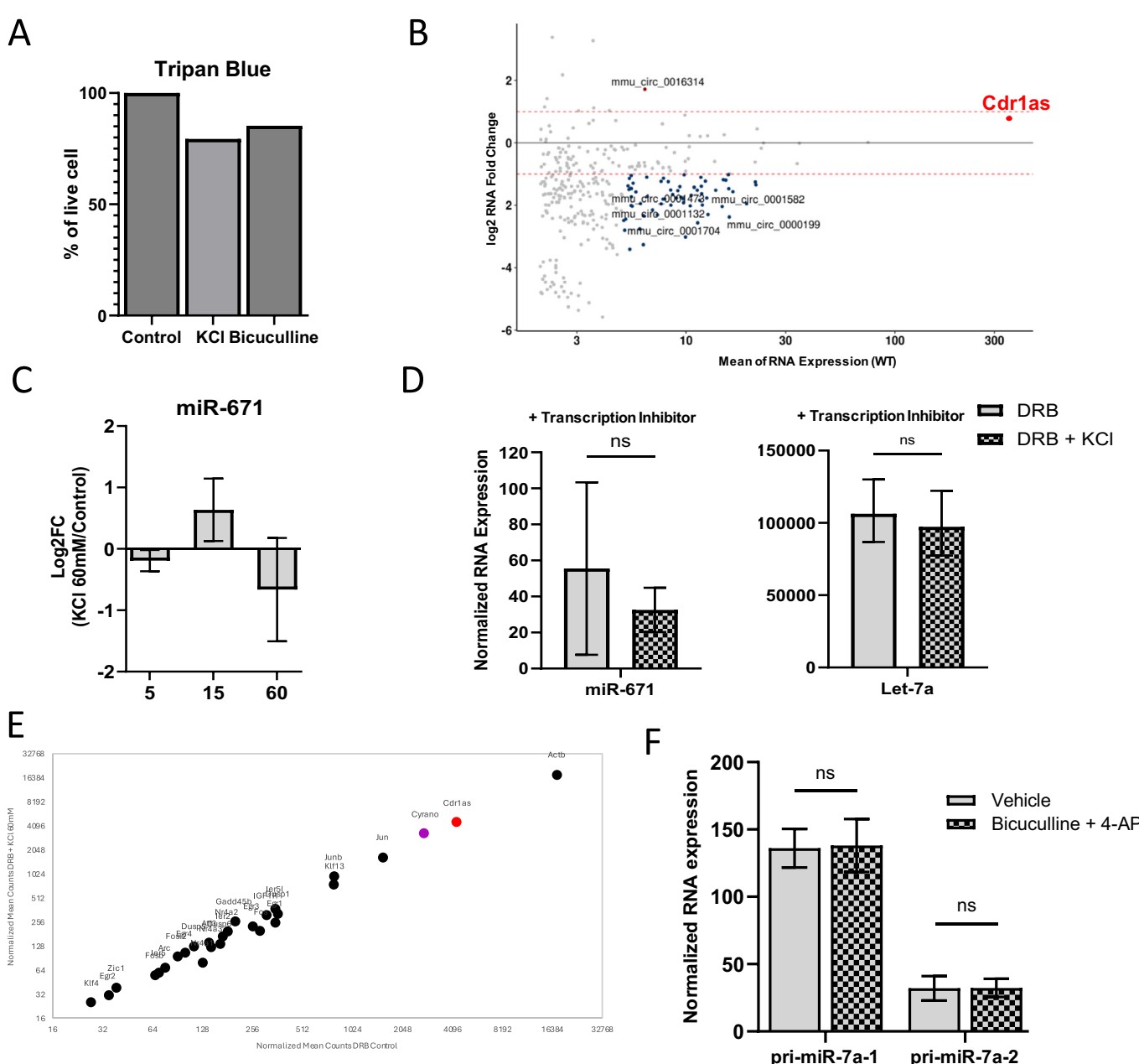

**Figure EV1. Sustained neuronal depolarization in neuronal cultures and transcription inhibitor controls.**

(A) Percentage of live cells after treatments with 60 mM KCl for 1 h (middle bar) or Biculine + 4-AP durin 30 min. (right bar), compare with untreated cells as negative control (left bar). Percentage of live cells was stimated based on Tripan blue staining after cell dissociation ("Methods"). (B) circRNA expression changes for WT neurons before and after K+ treatment. Plotted is the mean change of 2 independent biological replicates per condition. Red dot: Cdr1as. Blue and brown dots: other statistically significant circRNAs ("Methods"). (C) Expression changes of mature miR-671 quantified by TaqMan assay ("Methods"), after 5, 15 and 60 min of sustained KCl depolarization. Bar plot represents the mean of 3 biological replicates (3 independent primary cultures from 3 animals). *P* value: Mann–Whitney U test. Error: SD. (D) Expression levels of mature miR-671 (left) and let-7a (right) quantified by small RNA-seq for 3 independent primary cultures, before and after sustained depolarization plus pre-incubation with transcription inhibitor (DRB). *P* value: Mann–Whitney *U* test. Error: SD. (E) RNA quantification of IEGs after sustained KCl depolarization plus pre-incubation with transcription inhibitor (DRB) (Nanostring nCounter, "Methods"). RNA counts are normalized to housekeeping genes (Actb, Tubb5 and Vinculin). Cdr1as and Cyrano shown in red and purple, respectively. Each dot represents the mean of 3 biological replicates (3 independent primary cultures from 3 animals). (F) Pri-miR-7a-1 (left) and pri-miR-7a-2 (right) RNA expression changes after 30 min incubation with 50 μM Bicuculline + 75 μM 4-Aminopyridine (antagonist of GABA-A receptors and K+ Channel blocker, respectively), compared to corresponding vehicle control (DMSO, gray bars. RNA measured by Nanostring nCounter ("Methods"). RNA counts are normalized to housekeeping genes (Actb, Tubb5 and Vinculin). Bar plot represents the mean of 3 biological replicates (3 independent primary cultures from 3 animals). *P* value: Mann–Whitney *U* test. Error: SD.

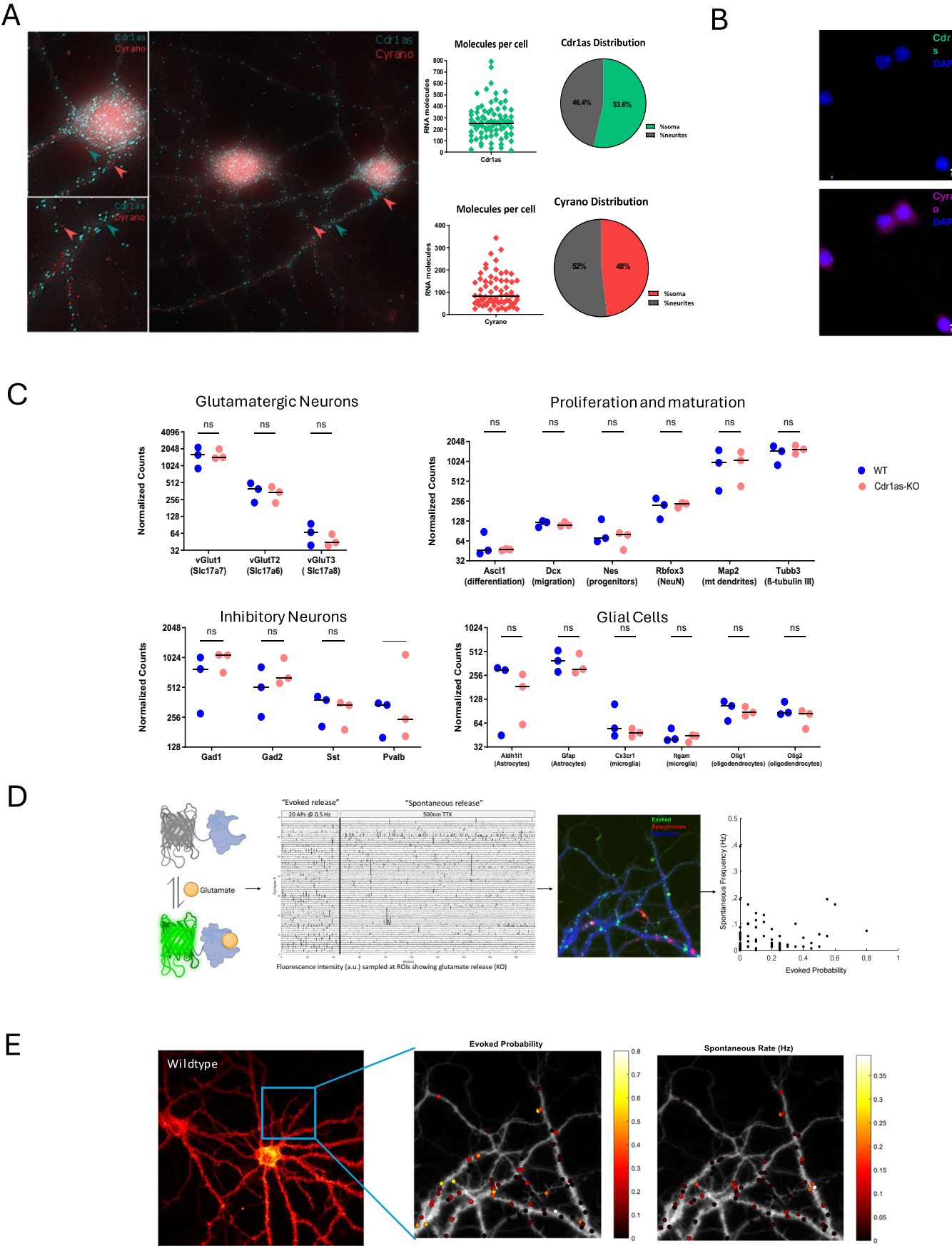

**Figure EV2. Characterization of cortical primary neuronal cultures.**

(A) Right: Single-molecule RNA FISH (Stellaris, "Methods") of Cdr1as (cyan) and Cyrano (magenta) performed in WT neurons. Left: smRNA FISH quantification of Cdr1as molecules (following Raj et al, 2008, "Methods"). Each dot represents the mean number of molecules in an independent cell (soma + neurites) (Cdr1as $n = 6$; 80 cells) (Cyrano $n = 5$; 69 cells). Horizontal line: Median. Pie charts show molecule distribution in somas versus neurites. (B) Control of Cdr1as probe specificity by single-molecule RNA FISH (Stellaris, "Methods") of Cdr1as (cyan) and Cyrano (magenta) Cdr1as-KO neurons DIV21. DAPI: blue. (C) Quantification of cellular markers to characterize WT versus Cdr1as-KO primary cultures DIV21 (Nanostring nCounter, "Methods"). RNA counts are normalized to housekeeping genes (Actb, Tubb5 and Vinculin). Each dot represents an independent biological replicate (3 independent primary cultures from 3 animals). Excitatory neurons (Sclc17a7, Sclc17a6, Sclc17a8), Inhibitory neurons (*Gad1, Gad2, Sst, Pvalb*), Proliferation and maturation markers (*Ascl1, Dxc, Nes, Rbfox3, Map2, Tubb3*) and Glial cells markers (*Aldh1, Gfap, Cx3cr1, Itgam, Olig1, Olig2*) are plotted. *P* value: Mann–Whitney U test. Horizontal bar: Median. (D) Transduction of a glutamate sensor (AAV, "Methods") into WT and Cdr1as-KO primary neurons followed by real-time imaging of excitatory synaptic terminals during AP-evoked (20 APs at 0.5 Hz) and spontaneous (5 min + 500 nM TTX) release conditions. (E) Visualization of glutamate sensor expression (GlusnFR) in WT primary neuron DIV20 (representative image). Region of interest zoom-in (yellow box). Middle and right panels indicate selection of active synaptic terminals and quantification of evoked probability and spontaneous frequency, respectively.

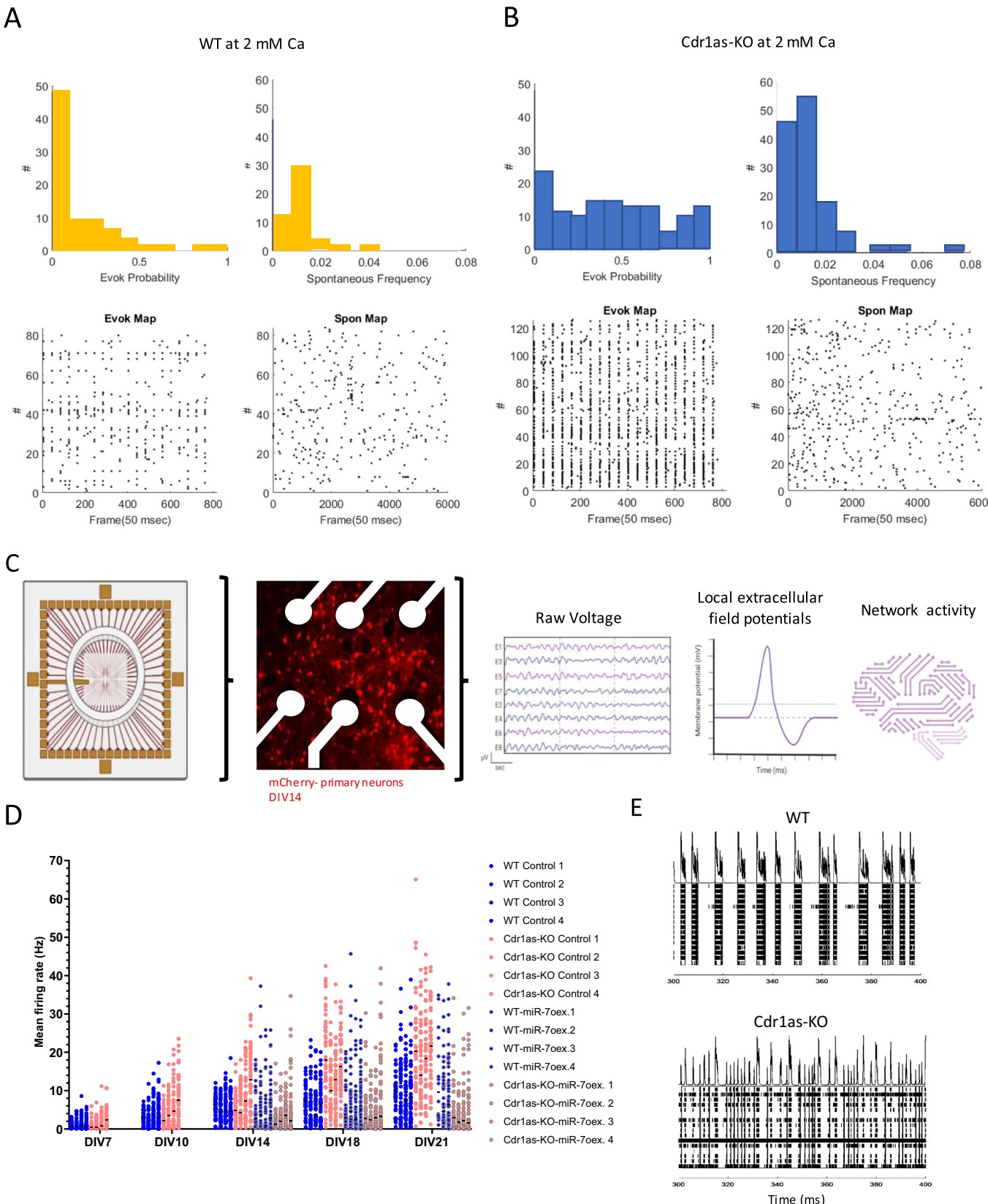

**Figure EV3. Glutamate release and multielectrode array (MEA) method and quality controls.**

(A) Histogram and scatter plot show distribution of number of active synapses of WT neurons cultured in media with 2 mM $Ca^{+2}$ to ensure neuronal firing. AP-evoked glutamate release calculated as evoked probability (left) and spontaneous glutamate release calculated as spontaneous frequency (right), respectively. (B) Histogram and scatter plot show distribution of number of active synapses of Cdr1as-KO neurons cultured in media with 2 mM $Ca^{+2}$ to ensure neuronal firing. AP-evoked glutamate release calculated as evoked probability (left) and spontaneous glutamate release calculated as spontaneous frequency (right), respectively. (C) Scheme of the Multi-electrode Array recording protocol (Axion Biosystems, CytoView MEA 48, "Methods"). Second panel: representative image of cultured neurons DIV14 in a recording well. white: electrodes, red: mCherry reporter. Third panel: schematic representation of output data, extracellular field potentials and neuronal network activity. AP: Adaptative threshold 6 SD. Sampling frequency 12.5 kHz. Active electrode selection criteria 5 spikes/minute. (D) Mean Firing Rate: Total number of spikes per single electrode divided by the duration of the analysis (600 s), in Hz. Each column represents recordings from a single biological replicate. Each dot represents recordings from a single electrode. (E) Example raw spikes from multi electrodes Array recording: 100 ms raw spikes of WT and Cdr1as-KO neurons DIV21. Each row represents one independent electrode (black bars).

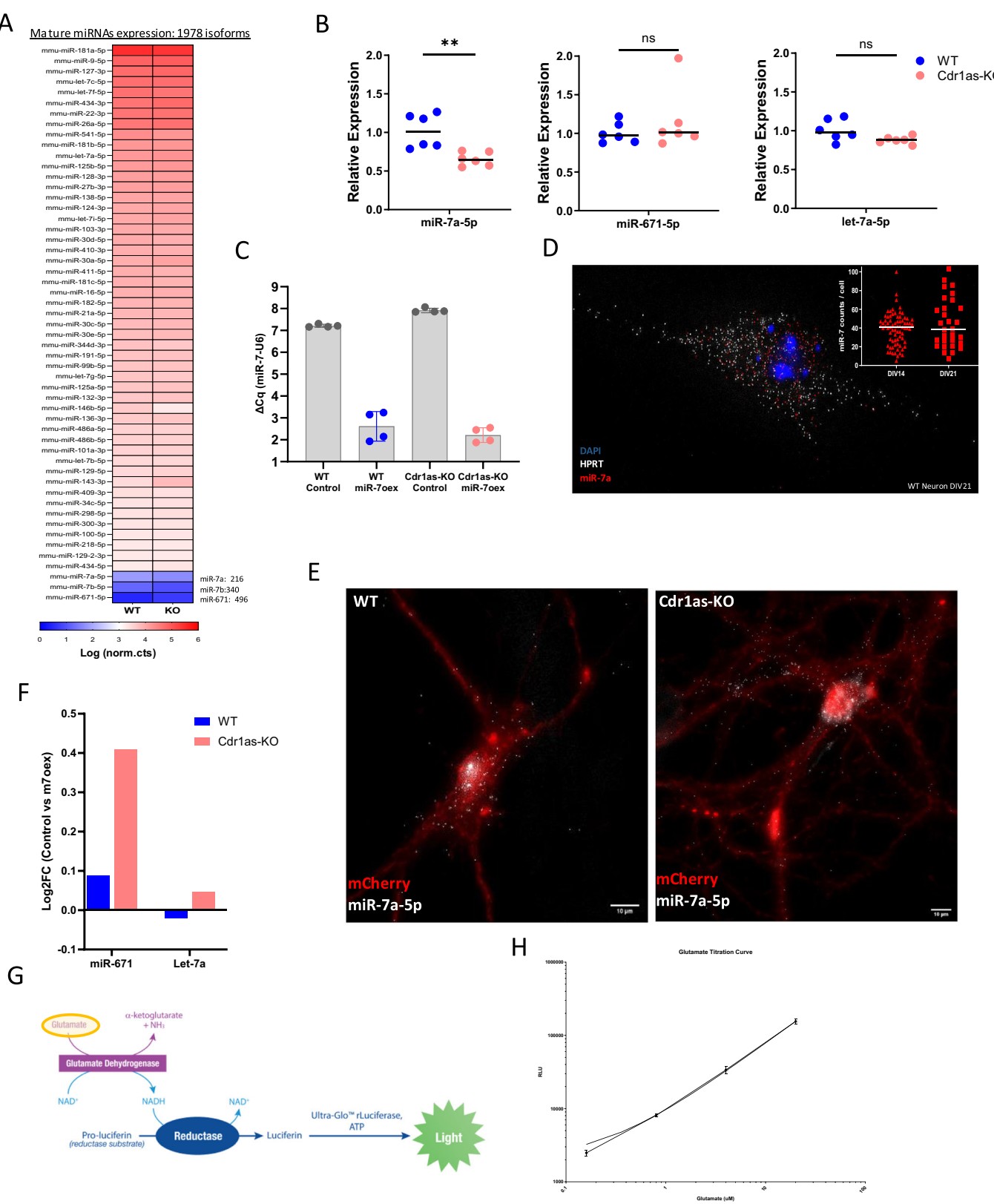

◀ **Figure EV4.  miRNAs in WT and Cdr1as-KO cortical neurons.**

(A) Heatmap from bulk smRNA-Seq ("Methods") (1978 miRNA isoforms detected), normalized expression plotted (log.norm.counts) of top 50 mature miRNAs in WT and Cdr1as-KO primary neurons DIV21, plus miR-7a, miR-7b and miR-671. (B) Quantification of mature miR-7a-5p, mir-671-5p and let-7a-5p (TaqMan Assay, "Methods") to characterize WT versus Cdr1as-KO primary cultures DIV21. RNA normalized to housekeeping genes (snRNA U6, snoRNA202) Each dot represents an independent biological replicate (6 independent primary cultures from 6 animals). *P* value: Mann–Whitney *U* test. Horizontal line: Median. (C) Quantification of mature miR-7a-5p (TaqMan Assay, "Methods") in WT and Cdr1as-KO primary cultures at 14 days post miR-7 overexpression. RNA normalized to housekeeping gene snRNA U6. Each dot represents an independent biological replicate (4 independent primary cultures from 4 animals). (D) Single-molecule RNA FISH (ViewRNA Plus, "Methods") of miR-7a-5p (red) and housekeeping gene Hprt (white) performed in WT neurons DIV14 and DIV21, DAPI: blue. Insert: smRNA FISH quantification of miR-7 molecules (following Raj et al, 2008, "Methods"). Each dot represents the mean number of molecules in an independent cell: DIV14 ($n = 75$ cells) and DIV21 ($n = 30$ cells). (Widefield microscopy 60×). (E) Single-molecule miRNA in situ hybridization (ViewRNA Plus, "Methods") of miR-7a-5p (white) and infection reporter mCherry (red), performed in WT and Cdr1as-KO neurons DIV21, 14 days post miR-7 overexpression (Widefield microscopy 60×). (F) Quantification of miR-671 and let-7a after miR-7 overexpression (14dpi) by RNA-Seq ("Methods") in WT and Cdr1as-KO neurons at DIV21. Bar plots represents mean of 4 independent biological replicates per genotype. (G) Schematic representation of the enzymatic principle behind glutamate secretion assay. Modified from Glutamate-Glo™ Assay ("Methods"). GDH enzyme catalyzes the oxidation of glutamate with associated reduction of NAD+ to NADH. In the presence of NADH, Reductase enzymatically reduces a pro-luciferin to luciferin. Luciferin using Ultra-Glo™ Luciferase and ATP, and the amount of light produced (RLU) is proportional to the amount of glutamate in the sample. (H) Standard curve for calibration of glutamate secretion assay. Serial dilutions curve of 50 µM Glutamate stock solution. Quantification of secreted glutamate concentrations for tested samples based on interpolation of RLU values.

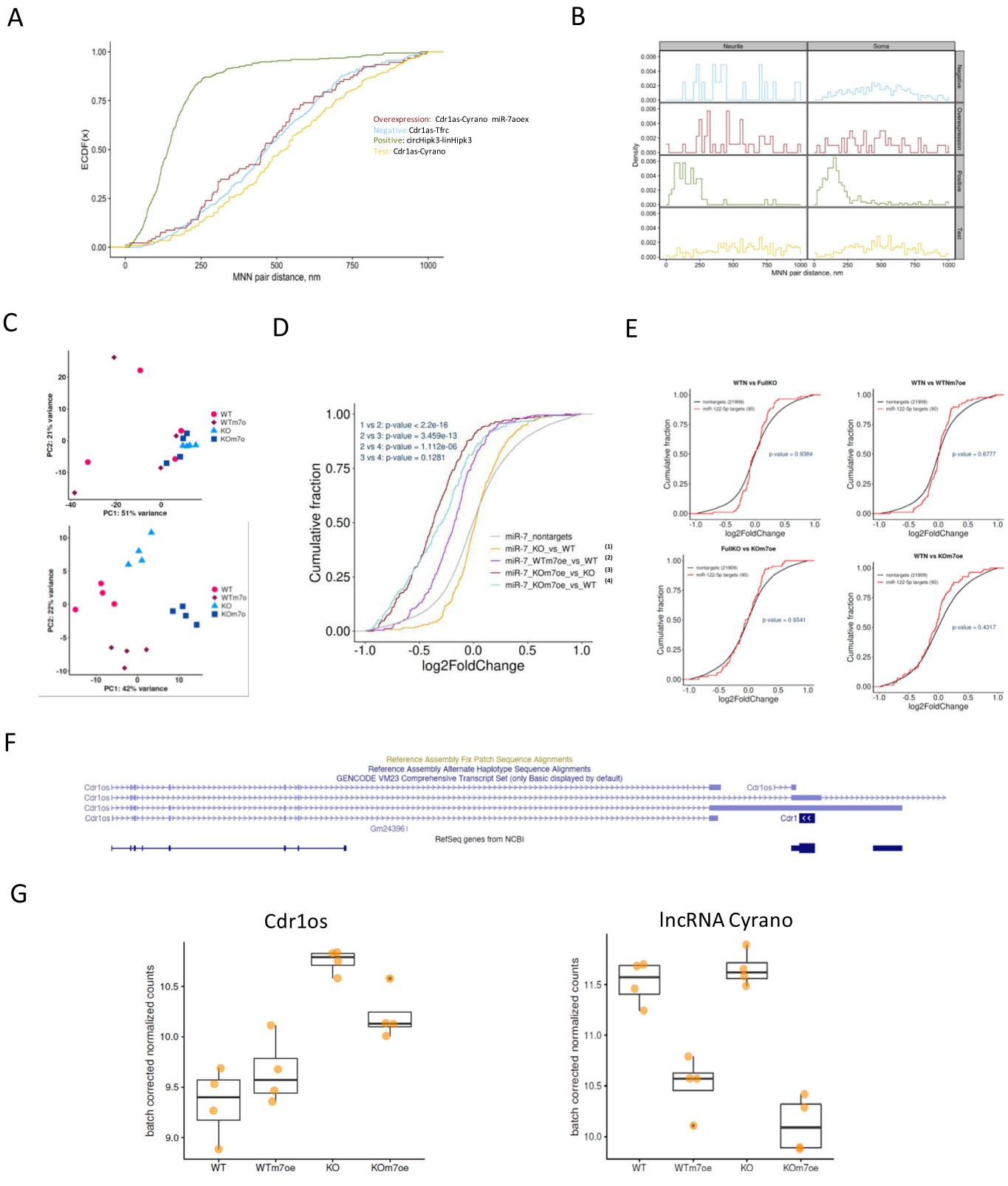

◄ **Figure EV5. Cdr1as and Cyrano colocalization analysis. miR-7 everexpression mRNA-sequencing quality controls.**

(A) Cumulative distribution function (CDF) plot of molecule distances based on smRNA FISH images of WT neurons DIV21. before and after miR-7 overexpression ("Methods") comparing all computationally predicted mutual nearest neighbor distances in nm all conditions (MNN, "Methods"). Positive technical control: circHipk3-linHipk3; Negative control: Cdr1as-Tfrc; Test: Cdr1as-Cyrano; overexpression: Cdr1as-Cyrano miR-7aoex. 2 independent biological replicates from 2 animals. (B) Density plot molecule distances based on smRNA FISH to compare somas versus neurites for all tested conditions. Analysis conditions same as in (A). (C) Principal component analysis (PCA) of all data sets. Each replicate represented by one dot. Original (up) and batch-corrected with nested design (down). (D) Cumulative distribution function (CDF) plot of gene expression comparing all computationally predicted miR-7 targets ("Methods") to mRNAs lacking predicted miR-7 target sites (non-targets, gray), across 4 independent biological replicates of WT, Cdr1as-KO, WT + miR-7 overexpression, Cdr1as-KO + miR-7 overexpression. *P* value: *U* Mann–Whitney test. (E) Cumulative distribution function (CDF) plot of gene expression comparing all computationally predicted miR-122 targets ("Methods", red) to mRNAs lacking predicted miR-7 target sites (non-targets, black), across 4 independent biological replicates of WT, Cdr1as-KO, WT + miR-7 overexpression, Cdr1as-KO + miR-7 overexpression. *P* value: *U* Mann–Whitney test. (F) Cdr1os transcript (Cdr1as precursor transcriptional unit) reference sequence alignments from Genome Browser (GENCODE VM23). (G) Gene expression of Cdr1os and lncRNA Cyrano in each dataset. Box plots, 4 independent biological replicates per condition, for each comparison tested. (FDR < 0.05).

