## [Peer Review File · EMBO Reports]

miR-7 controls glutamatergic transmission and neuronal connectivity in a Cdr1as dependent manner

Cledi A. Cerda Jara, Seung Joon Kim, Gwendolin Thomas, Zohreh Farsi, Grygoriy Zolotarov, Giuliana Dube, Aylina Deter, Ella Bahry, Elisabeth Georgii, Andrew Woehler, Monika Piwecka, and Nikolaus Rajewsky

Corresponding author(s): Nikolaus Rajewsky (rajewsky@mdc-berlin.de)

Review Timeline:

Submission Date:	4th Jan 24
Editorial Decision:	18th Jan 24
Revision Received:	23rd Mar 24
Editorial Decision:	10th Apr 24
Revision Received:	12th Apr 24
Accepted:	14th May 24

Editor: Esther Schnapp

Transaction Report: Please note that the manuscript was transferred from another journal where it was originally reviewed. Since the original reviews are not subject to EMBO's transparent review process policy, they cannot be published.

Dear Nikolaus,

Thank you for the transfer of your revised manuscript to EMBO reports. I contacted an advisor, who turned out to be referee 1 at Cell reports.

S/he is overall fine with your manuscript and supports its acceptance, except that the concerns regarding Fig 4A-D have not been satisfactorily addressed:

"From the technical side, the authors addressed most of my concerns raised in my review for Cell Reports (I was reviewer 1). However, my concerns regarding Fig. 4A-D (smFISH of Cdr1as and Cyrano) are still valid. The representative pictures are not convincing (especially for the Cyrano condition, Fig. 4C), and the statistical analysis is not appropriate (lumping together neurites from only two independent experiments, without correcting for potential batch effects etc.)."

I would therefore like to invite you to provide better images for Fig 4A-D and to correct the statistical analysis. The magnifications of the images in 4A and in 4C should also be equal for a better comparison. Please let me know if you have any comments or questions, and we can discuss the revisions further, also in a video chat, if you like.

In addition to addressing the referee comments, the following editorial requests will need to be addressed.

Please provide with your final ms:

1) a .docx formatted version of the manuscript text (including legends for main figures, EV figures and tables). Please make sure that the changes are highlighted to be clearly visible. The manuscript sections should be in the following order: Title page - Abstract & Keywords - Introduction - Results - Discussion - Materials & Methods - Data Availability - Acknowledgments - Disclosure Statement & Competing Interests - References - Figure Legends - Expanded View Figure Legends.

2) individual production quality figure files as .eps, .tif, .jpg (please upload one file per figure). See https://wol-prod-cdn.literatumonline.com/pb-assets/embo-site/EMBOPress_Figure_Guidelines_061115-1561436025777.pdf for more info on how to prepare your figures.

3) We replaced Supplementary Information with Expanded View (EV) Figures and Tables that are collapsible/expandable online. A maximum of 5 EV Figures can be typeset. EV Figures should be cited as "Figure EV1, Figure EV2" etc... in the text and their respective legends should be included in the main ms text after the legends of main ms figures.

- For the figures that you do NOT wish to display as Expanded View figures, they should be bundled together with their legends after each figure in a single PDF file called *Appendix*, which should start with a short Table of Content with page numbers. Appendix figures should be referred to in the main text as: "Appendix Figure S1, Appendix Figure S2" etc. See detailed instructions regarding expanded view here: <https://www.embopress.org/page/journal/14693178/authorguide#expandedview>

5) a complete author checklist, which you can download from our author guidelines <https://www.embopress.org/page/journal/14693178/authorguide>. Please insert information in the checklist that is also reflected in the manuscript. The completed author checklist will also be part of the RPF.

6) Before submitting your revision, primary datasets produced in this study need to be deposited in an appropriate public database (see <https://www.embopress.org/page/journal/14693178/authorguide#datadeposition>). Please remember to provide a reviewer password if the datasets are not yet public. The accession numbers and database should be listed in a formal "Data Availability" section placed after Materials & Method (see also <https://www.embopress.org/page/journal/14693178/authorguide#datadeposition>). Please note that the Data Availability Section is restricted to new primary data that are part of this study. * Note - All links should resolve to a page where the data can be accessed. *

7) At EMBO Press we ask authors to provide source data for the main manuscript figures. Our source data coordinator will contact you to discuss which figure panels we would need source data for and will also provide you with helpful tips on how to

upload and organize the files.

8) Our journal also encourages inclusion of *data citations in the reference list* to directly cite datasets that were re-used and obtained from public databases. Data citations in the article text are distinct from normal bibliographical citations and should directly link to the database records from which the data can be accessed. In the main text, data citations are formatted as follows: "Data ref: Smith et al, 2001" or "Data ref: NCBI Sequence Read Archive PRJNA342805, 2017". In the Reference list, data citations must be labeled with "[DATASET]". A data reference must provide the database name, accession number/identifiers and a resolvable link to the landing page from which the data can be accessed at the end of the reference. Further instructions are available at <https://www.embopress.org/page/journal/14693178/authorguide#referencesformat>

9) Regarding data quantification (see Figure Legends:

<https://www.embopress.org/page/journal/14693178/authorguide#figureformat>)

- the name of the statistical test used to generate error bars and P values,
- the number (n) of independent experiments (please specify technical or biological replicates) underlying each data point,
- the nature of the bars and error bars (s.d., s.e.m.),
- If the data are obtained from $n < 3$, use scatter blots showing the individual data points.

Currently missing:

- Please define the annotated p values ***/**/* in the legend of figure 2b-c; 3a-f; 4b, d-f; as appropriate.
- Please indicate the statistical test used for data analysis in the legend of figure 2e.
- Please note that the error bars are not defined in the legend of figure 2e.

10) Please add up to 5 keywords to the ms file.

11) Please add a "Disclosure and Competing Interest Statement" to the ms file.

12) Please remove the author credits from the ms file. All credits need to be entered online during ms submission.

13) Some funding info is missing in our online system, please add.

14) FIGURE CALLOUTS: missing- Fig. 1A; Suppl. Figures 1-3 called out before Fig. 1G - please try to call out the main figures and their panels in a consecutive order; Fig. 5F called out but panel F doesn't exist in this figure; Suppl. Tables 1-6 called out but are missing in the submission.

15) The abstract needs to be written in present tense when you describe the new data. Please correct.

16) EMBO press papers are accompanied online by A) a short (1-2 sentences) summary of the findings and their significance, B) 2-3 bullet points highlighting key results and C) a synopsis image that is exactly 550 pixels wide and 200-600 pixels high (the height is variable). You can either show a model or key data in the synopsis image. Please note that text needs to be readable at the final size. Please send us this information along with the final manuscript.

As part of the EMBO publication's Transparent Editorial Process, EMBO reports publishes online a Review Process File (RPF) to accompany accepted manuscripts. This File will be published in conjunction with your paper and will include the referee reports, your point-by-point response and all pertinent correspondence relating to the manuscript. In your case, we only have referee 1's comments and your reply.

I look forward to seeing a final version of your manuscript when it is ready.

Dear Dr. Schnapp,

Thank you very much for your consideration of our transferred manuscript and for your diligent work. We are very glad to see that the reviewer supports the acceptance of our manuscript.

Regarding their concern about figure 4A-D, we have carefully considered all the technical issues raised by the referee and below you can find:

- (1) Improved quality of representative images for Fig. 4A-D, with corrected magnifications.
- (2) Improved statistical analysis of the images, separated by animals, cell compartments, and neuronal replicates.
- (3) Adapted descriptions and conclusions from smFISH and quantification data, according to the new improved analysis, as suggested previously from the reviewer.

Reviewers' comment is shown here in **black**, our responses, in **blue**.

Reviewers' comments:

"From the technical side, the authors addressed most of my concerns raised in my review for Cell Reports (I was reviewer 1). However, my concerns regarding Fig. 4A-D (smFISH of Cdr1as and Cyrano) are still valid.

The representative pictures are not convincing (especially for the Cyrano condition, Fig. 4C), and the statistical analysis is not appropriate (lumping together neurites from only two independent experiments, without correcting for potential batch effects etc.)."

We understand the concern of the referee regarding the processed smFISH images shown as representative examples in Figure 4. To address this matter, we did the following changes:

(1) re-analyzed and re-processed all used smFISH images starting from the original raw imaging data. We increased the pixel depth, inverted the images for better contrast and accentuated the resolution of the spots, for each fluorophore independently (Quasar 670: Cdr1as and Quasar 570). We believe we have now improved the clarity and sharpness of signals for Cdr1as (NEW Figure 4A and B) and Cyrano (NEW Appendix Figure S4A and B) RNAs.

Additionally, we have added zoom in inserts of somato-neuritic compartments that show more details of the molecule distributions for each condition and especially the radical change observed in Cdr1as expression upon miR-7 overexpression. As shown here below:

Regarding the concern about the quality of smFISH for Cyrano, we want to address the fact that Quasar 570 fluorophore, the one combined with Cyrano probe, has some autofluorescence background on its own despite the probe coupled to it; also considering the imaging was done with widefield microscopy (Nikon Ti-E microscope and Andor iXON Ultra DU-888 camera; Methods) some of the finest background

noise cannot be removed. Therefore, the images might appear not as sharp or the dots not as clear as for Cdr1as.

Therefore, we have now adapted the main article accordingly. We removed any conclusions drawn by smFISH-data-only about Cyrano distribution changes in the neurons. We have moved Cyrano smFISH data to supplementary figures, as we think it is not central to our main conclusions and does not change the key discoveries of our manuscript regarding the relation between Cdr1as and miR-7.

We have now only used the observations about Cyrano changes in gene expression, after miR-7 upregulation, obtained by independent quantifications of RNA by Nanostring or bulk RNA-sequencing to conjecture about the possible biological mechanisms acting behind miR-7 and Cdr1as interaction (**lines 420-426**).

(2) Regarding the inappropriate statistical analysis of the smFISH data. The referee was concerned about the lumping of neurites and somas, and the disregard of potential batch effects between independent animals.

To solve this, we have removed the lumped plots and reanalyzed, and we added the number of Cdr1as molecules normalized by area of each independent biological separately (2 animals), in each soma and each neurite of each neuron used for quantification analysis (**New Figure 4C-D**). Addressing the separated analyses in the main manuscript as well (**lines 412-413**). Additionally, we move to the appropriate nonparametric and unpaired statistical test (U Mann Whitney) to control abnormal distributions.

Finally, regarding the potential batch effect among independent experiments. Although the absolute number of molecules per area might differ from animal to animal, **we did not observe any batch effect between the independent experiments**. On the contrary, we observe the same significant differences in both animals regarding molecule distribution, for somas and neurites equally. As we show here below and in **NEW Figure 4C-D**.

(3) We adapted the main text in the manuscript according to the new conclusions of the smFISH data. Mainly we removed any hypothesis regarding a exclusive mechanism of removal of Cdr1as in neurites, cause the down-regulation of Cdr1as molecules after miR-7 overexpression seems more of a global effect in soma and neurite compartments **(Lines 410-415, lines 442-445 and New Figure 4C-D)**.

The conclusions about the distribution of Cyrano, the counterpart of Cdr1as in miR-7 regulation, have also been adapted. The smFISH data alone showed no evident difference in Cyrano's compartment distribution, compared to the striking downregulation of Cdr1as. Therefore, we took only global observations about Cyrano gene expression changes, after miR-7 upregulation, obtained by independent quantifications of global RNA using Nano string and bulk RNA-sequencing. These observations anyhow allowed us to hypothesize about the possible biological mechanisms acting behind miR-7 and Cdr1as interaction **(lines 442-445) and lines 699-702)**.

All corresponding changes on the main manuscript mentioned above, are shown in **orange** in the main text (including previous rounds of revisions).

Dear Nikolaus,

Thank you for the submission of your revised manuscript. Referee 1 now supports its publication and we can therefore in principle accept it.

There are only still a few issues with the ms files that need to be sorted:

- 1 keyword needs to be deleted, I suggest deleting "plasticity".
- The DAS provides a link and a token, BUT before publication, so within 2 weeks from acceptance, we need a direct link that resolves to the dataset. Do you may be have a direct link already?
- The source data (SD) for each figure need to be provided as separate and clearly labeled files which are zipped up into one folder so that we have one zip folder per one figure; each file in the folder needs to have the panel label/name. Currently SD for Figure 2 and 3 do not have clearly labeled panels while the separately uploaded files/folders for Figures 4 and 5 need to be grouped into separate Figure 4 and 5 folders.
- Some funding info is missing in our online ms submission system: EU ITN - circular RNA Biology Training Network: circRTrain (721890), Helmholtz Association's Initiative and Networking Fund, Polish National Agency for Academic Exchange (Polish Returns grant no. PPN/PPO/2019/1/00035/U/0001) and the National Science Centre (grant no. 2018/30/E/NZ3/00624). Please add this info, as all funding info need to be in the ms file and in our online submission system.
- The last bullet point is not clear to me:
 - miR-7 up-regulation causes a synergetic gene regulation exerted by Cdr1as:miR-7 interaction and is sufficient to regulate local and network neural activity and glutamate secretion... What is "synergetic gene regulation"? Can the verb "regulate" be replaced by something more specific?
- We still need a synopsis image from you that is exactly 550 pixels wide and 200-600 pixels high (the height is variable). You can either show a model or key data in the synopsis image. Please note that text in the image needs to be readable at the final image size.

When you start a new ms version in our system, you can bring forward all old files and only replace the files that need to be replaced, which should hopefully make it easier.

Referee #1:

The authors have now addressed my remaining concerns regarding the presentation and analysis of the single-molecule FISH data. I recommend publication.

The authors have addressed all minor editorial requests.

Nikolaus Rajewsky
Max Delbrueck Centrum fuer Molekulare Medizin
Robert-Roessle-Str. 10
Berlin-Buch, Berlin 13125
Germany

Dear Dr. Rajewsky,

I am very pleased to accept your manuscript for publication in the next available issue of EMBO reports. Thank you for your contribution to our journal.

Yours sincerely,
